# New Directions in Modeling and Computational Methods for Complex Mechanical Dynamical Systems

**Firdaus E. Udwadia** [1,2,*] **and Nami Mogharabin** [1]

1 Department of Aerospace and Mechanical Engineering, University of Southern California, Los Angeles, CA 90089, USA
2 Department of Civil and Environmental Engineering, University of Southern California, Los Angeles, CA 90089, USA
* Correspondence: feuusc@gmail.com

**Abstract:** This paper presents a new conceptualization of complex nonlinear mechanical systems and develops new and novel computational methods for determining their response to given applied forces and torques. The new conceptualization uses the idea of including particles of zero mass to describe the dynamics of such systems. This leads to simplifications in the development of their equations of motion and engenders a straightforward new computational approach to simulate their behavior. The purpose of the paper is to develop a new analytical and computational methodology to handle complex systems and to illustrate it through the study of an old unsolved problem in classical mechanics, that of a non-uniform rigid spherical shell rolling, without slipping, under gravity on an arbitrary dimpled bowl-shaped rigid surface. The new conceptualization provides the explicit equations of motion for the system, the analytical determination of the reaction forces supplied by the surface, and a straightforward computational approach to simulate the dynamics. Detailed analytical and numerical results are provided. The computations illustrate the complexity of the dynamical behavior of the system and its high sensitivity to the initial orientation of the shell and to the presence of any initial angular velocity normal to the surface.

**Keywords:** new approaches to nonlinear and multibody dynamics; constrained motion; zero-mass particles; shell rolling on an arbitrary surface; computational approaches

## 1. Introduction

In the area of dynamical systems, mechanical systems play a quintessential role. One of the first mathematical descriptions of a dynamical system was a mechanical one as proposed by Newton in his description of the motion of a particle subjected to forces. Euler gave the equations of the motion of particles and rigid bodies in 1752 [1,2]. Numerous advances in the general theory of dynamical systems have been initiated and/or brought about by investigations into the behavior of mechanical systems. For example, the discovery of chaos in dynamical systems is credited to Poincaré for his geometrical analysis of the three-body planetary problem [3,4], and the general theory of bifurcations in dynamical systems was presaged by observations on the mechanical buckling of columns. In fact, as is well known, the modern geometrical approach currently used in the analysis of nonlinear dynamical systems has its roots in the approach pioneered by Poincaré in his essay on the three-body problem [4].

This paper deals with modeling and computational aspects of nonlinear multi-body mechanical systems. Most mechanical systems are nonlinear, even simple ones such as planar pendulums. Thus, the response of mechanical multi-body systems that include particles, solid, and/or flexible bodies subjected to externally applied actions can indeed be both complex and highly nonlinear. This paper presents a new conceptualization and methodology to model complex multi-body systems to obtain their response to the time-dependent forces and torques to which they may be subjected.

Each multi-body mechanical system, depending on its number of degrees of freedom, needs a certain minimum number of time-dependent variables (coordinates, parameters) to model it mathematically. These variables describe the motion of the system in the so-called configuration space. The aim of obtaining a suitable model is to: (1) write down expressions for the second derivatives of these coordinates with respect to time in terms of the coordinates, their first derivatives, time, and the given time-dependent actions, such as forces and torques, acting on the system, and (2) corroborate the consequent model's predicted behavior of interest with what is physically observed.

One of the major areas of considerable concern in the development of the dynamical equations of motion for multi-body systems is the handling of constraints. From a historical perspective, this concern has been one of the central foci of attention of mathematicians/mechanicians throughout the last 300 years or so [5–7]. It is only recently that advances have been made in the development of a general theory of constrained mechanical systems that can be applied with equal ease to both holonomic and non-holonomic constraints [8–10]. Furthermore, it is now also possible to include the dissipation and/or injection of energy to the system through the constraints, as often happens in physical situations [11,12]. These advances expand our understanding of the nature of constraints, their use, and their influence on a system when they are imposed on it.

In this expanded view, constraints can be used: (1) for modeling restrictions applied to mechanical systems so that they satisfy certain kinematical conditions during their motion, such as those imposed on the relative motion between robots joints [10]; (2) for finding the explicit actions (forces and torques) needed for a mechanical system to follow a given trajectory, such as requiring the end effector of a robot arm to move exactly along a prescribed trajectory, or requiring the system to satisfy some dynamical conditions [13]; (3) for obtaining the equations of motion of a system when additional coordinates are required beyond the minimum, such as in the determination of the rotational motion of a rigid body whose orientation though described by just three coordinates—the Euler angles—requires, in fact, four parameters—a quaternion—to describe it adequately and thereby avoid the singularities in angular velocities that arise when using Euler angles [14].

In addition to the above uses, complex multi-body systems often have numerous constraints to which they are subjected, and obtaining the equations of motion for such systems when using the minimum (though adequate) number of coordinates can not only be very time-consuming but, more importantly, can lead to equations of high complexity. This makes understanding the behavior of such nonlinear systems much more difficult and also usually results in higher computational complexity. The complexity in formulating equations of motion arises because the numerous constraints that the system is required to respect may not be easily describable in terms of the minimum adequate number of chosen coordinates. Often, there may be some other time-dependent coordinates, besides the minimum adequate number chosen, that simplify the description of the constraint relations and/or the description of the unconstrained system, for instance, the coordinates of contact points, joints, coordinates on which the external forces/torques act, or other significant points in the system that aid our geometrical understanding of its behavior.

The new and novel direction proposed in this paper is the direct inclusion of additional time-dependent variables (coordinates) into the dynamical description of a multi-body system in order to make its unconstrained motion simpler to obtain and/or the constraints simpler to express mathematically. On the face of it, such an expansion of the number of dynamical variables—the configuration space—poses unsurmountable problems from a conceptual viewpoint, since the mass associated with these additional coordinates must necessarily be zero in order to maintain the fidelity of the model to the physical system, and the inclusion of particles of zero mass would lead to singular mass matrices. Furthermore, in nonrelativistic mechanics, no forces can be applied to a particle with zero mass and hence its acceleration becomes indeterminate.

However, there have been significant developments in deriving the equation of motion of constrained mechanical systems in recent years [15,16]. It is these advances that

view a mechanical system from a different perspective as a system moving under constraints that makes the inclusion of zero-mass particles in the description of a mechanical system possible.

This paper illustrates the new modeling methodology and its associated computational aspects by solving, as an example, one of the unsolved problems of classical mechanics: the dynamics of a rigid non-uniform spherical shell rolling under gravity, without slipping, on an arbitrarily given surface. The dynamics of a rigid non-uniform sphere rolling on a surface under gravity is a classical problem that appears to have been first broached by E. Lindelöf in 1895 and further explicated by Chaplygin [17–19]. Kilin [20] investigated the trajectory of the contact point between a rolling sphere and a plane by deriving the equations of the motion. Borisov et al. [21–24] used one more non-holonomic constraint on spinning and studied the motion of a rigid body rolling with or without spinning on a plane. Bizyaev et al. [25] extended this topic by investigating the motion of a rolling spherical shell with a moving rigid body inside it. Ivanov has considered the long-time asymptotic motion of a homogeneous sphere moving on a horizontal rough plane with friction [26]. Besides motion on a plane, work on the motion of a non-uniform sphere on some simple prescribed surfaces has also been investigated by several researchers (e.g., see Ref. [27]). An investigation into the controlled motion of a spherical robot on a periodically vibrating horizontal plane has been recently carried out in Ref. [28]. Despite the work done on this classical problem by several investigators, the determination of the explicit equations of motion and the quantitative characteristics of motion for general (arbitrarily prescribed) surfaces on which a non-uniform shell/sphere rolls has remained an unsolved problem.

The structure of this paper is as follows. Section 2 provides the analytical development of the methodology. Section 2.1 begins with modeling the unconstrained multi-body system that includes the rigid spherical shell subjected to gravity and two zero-mass particles. In Section 2.2 the constraints are described and the equations of constraint are obtained. Sections 2.1 and 2.2 demonstrate using the zero-mass particles eases deriving the equations. Sections 2.3 and 2.4 provide the explicit equations of motion for the system. The determination of the physical constraint forces, and from them, the (generalized) reaction forces provided by the surface during the motion of the shell, are explained in Section 2.5. Extensive computational results are provided in Section 3. They deal with the dynamical behavior of a non-uniform rigid shell moving over a multi-dimpled bowl-shaped rigid surface. Three examples are illustrated. In Section 3.1 the details of the three examples used in the numerical computations are explained along with the initial orientations of the shell that are considered. Section 3.1.1 presents results for the first example in which the shell is given no initial spin velocity about the normal to the surface. In Section 3.1.2 the second example is studied in which the shell's dynamical behavior is computed when an initial spin velocity is added. Section 3.1.3 considers the third example in which a shell is restricted throughout its motion to have no spin velocity at all. Section 4 provides a discussion of the new approach to modeling complex dynamical systems and summarizes the computational results. Section 5 gives the conclusions.

An initial study using a zero-mass particle to model a solid sphere rolling on an arbitrary surface was reported in Ref. [29]. In the present paper, this study is greatly expanded and further explored. It differs from our initial study in the following aspects.

(1) The methodology is developed for a non-uniform shell and two particles of zero mass are used instead of one. This greatly simplifies the formulation of the problem from that obtained in Ref. [29] and the formulation of the equations of motion and the equations of constraint. It also makes the computations more efficient.

(2) The theory of constrained motion with singular mass matrices is used to obtain the final equations of motion of the system [15,16,30,31]. This theory requires a certain condition to be satisfied in order to yield the correct equations of motion for the physical system. In Ref. [29] this condition was only computationally confirmed for the parameters chosen in the numerical example presented there. Here, we show that the condition is analytically satisfied, thereby placing the approach on a firm

mathematical footing. This allows us to obtain the explicit closed form equations of motion for the shell moving over an arbitrarily prescribed surface.

(3)　Computational results that show the motion of the shell on a complex unsymmetrical multi-dimpled bowl-shaped surface with an unsymmetrical cross-section are obtained, showing vast qualitative differences in its motion and sensitive dependence on initial conditions.

(4)　Analytical equations for the reaction of the surface to the motion of the shell from the determination of the generalized forces of constraint are explicitly obtained. That is, besides obtaining the coordinates that describe the configuration of the system at each time instant and the velocities of these coordinates as done in Ref. [28], the generalized forces acting on the spherical shell at its point of contact with the surface are also determined. Thus, the reaction forces exerted by the surface are therefore explicitly obtained. This permits the minimum coefficient of friction required to sustain the motion of the shell over the surface, without any slippage, to be determined.

(5)　The effects of the initial orientation and the initial spin velocity of the shell—the component of the initial angular velocity normal to the surface—are investigated in considerable detail, showing that they have a significant effect on its motion.

(6)　A further constraint that prevents the shell to have any spin velocity is investigated. Its effect on the motion of the shell, and especially on the reaction forces that it brings about, is investigated in some detail.

## 2. Analytical Results

### 2.1. Description of the Unconstrained Multi-Body System

Figure 1 shows the unconstrained system consisting of a non-uniform rigid spherical shell, $\Sigma$, with mass $m$ and (geometric) center $O$, whose internal and external radii are $r_i$ and $r$, respectively. Its center of mass, denoted by $C$, is located at $w_C = [x_C, y_C, z_C]^T$ in an inertial coordinate frame $XYZ$ (see Figure 1). The body-fixed coordinate frame $\hat{x}\hat{y}\hat{z}$ is a right-handed coordinate frame formed by the principal axes of inertia of $\Sigma$; its origin is the point $C$. The location of the point $C$ with respect to $O$ is given by $\rho = [\rho_1, \rho_2, \rho_3]^T$, in which $\rho_i$, i $= 1, 2, 3$ are the components of the vector $\vec{\rho}$ in the body-fixed $\hat{x}\hat{y}\hat{z}$ coordinate frame. The principal moments of inertia of $\Sigma$ about these body-fixed axes are denoted by $J_1$, $J_2$, and $J_3$; the inertia tensor of the shell in the body-fixed coordinate frame is $J = Diag(J_1, J_2, J_3)$.

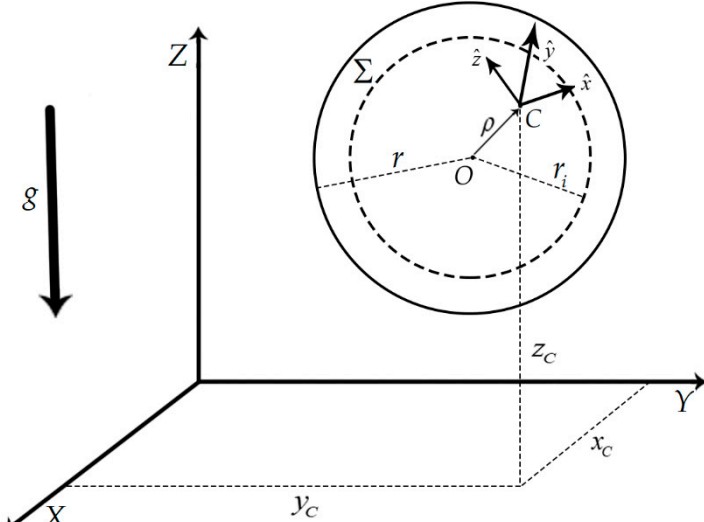

**Figure 1.** The unconstrained system consisting of the non-uniform spherical shell, $\Sigma$, with center of mass at $C$ and geometric center at $O$, under the force of gravity.

To describe the rotational motion of $\Sigma$, a four-vector quaternion (and not the Euler angles) is used; this averts singularities in determining the angular velocities. The

quaternion four-vector that describes the rotational configuration at any time $t$ is given by $u(t) := [u_0(t), u_1(t), u_2(t), u_3(t)]^T$, where the $u_i's$ are constrained to satisfy the relation $\|u(t)\|^2 = u_0(t)^2 + u_1(t)^2 + u_2(t)^2 + u_3(t)^2 = 1$ [14]. Thus, the position and orientation of the shell at any time $t$ can be described by the seven-vector $\tilde{q} := [x_C, y_C, z_C, u_0, u_1, u_2, u_3]^T = [w_C^T, u^T]^T$ wherein the *four*-vector $u$ satisfies the constraint $\|u(t)\| = 1$.

The unconstrained equation of motion of the shell $\Sigma$ under gravity can be obtained quite trivially using Lagrange's equation (see Figure 1). The total kinetic energy, $T$, of the shell is composed of its translational and rotational kinetic energy as $T = T_{trans} + T_{rot} = (1/2)m\dot{w}_C^T\dot{w}_C + (1/2)\omega^T J\omega$, where $\omega := [\omega_{\hat{x}}, \omega_{\hat{y}}, \omega_{\hat{z}}]^T$ is the absolute angular velocity of the shell whose components are measured in the body-fixed coordinate frame $\hat{x}\hat{y}\hat{z}$ with its origin at $C$. Dots over the symbols refer to derivatives with respect to time $t$ in this paper.

Since the absolute angular velocity of the body expressed in the body-fixed coordinate system $\hat{x}\hat{y}\hat{z}$ can be written in terms of the components of the quaternion four-vector $q$ as

$$\omega := \begin{bmatrix} \omega_{\hat{x}} \\ \omega_{\hat{y}} \\ \omega_{\hat{z}} \end{bmatrix} = 2E\dot{u} := 2\begin{bmatrix} -u_1 & u_0 & u_3 & -u_2 \\ -u_2 & -u_3 & u_0 & u_1 \\ -u_3 & u_2 & -u_1 & u_0 \end{bmatrix}\dot{u}, \tag{1}$$

the expression for the kinetic energy, $T$, can be written as

$$T(\dot{w}_C, u, \dot{u}) = T_{trans} + T_{rot} = \frac{m}{2}\dot{w}_C^T\dot{w}_C + 2\dot{u}^T E^T JE\dot{u} = \frac{m}{2}\dot{w}_C^T\dot{w}_C + 2u^T\dot{E}^T JE\dot{u}. \tag{2}$$

From the definition of $E$ in Equation (1) we see that the product $Eu = 0$, and on differentiating this relation with respect to time, $t$, it yields $E\dot{u} = -\dot{E}u$; this is used in the last equality in Equation (2). The potential energy of $\Sigma$ is given by

$$V(w_C) = mgz_C = mgw_C^Te_3, \tag{3}$$

where $e_3 = [0, 0, 1]^T$. The Lagrangian of the system is $L = T(w_C, u, \dot{u}) - V(w_C)$ and the equations of motion of the shell are simply given by

$$\frac{d}{dt}\left(\frac{\partial L}{\partial \dot{\tilde{q}}}\right) - \frac{\partial L}{\partial \tilde{q}} = 0. \tag{4}$$

Noting that

$$\frac{\partial T}{\partial \dot{u}} = 4E^T JE\dot{u}, \tag{5}$$

and

$$\frac{d}{dt}\left(\frac{\partial T}{\partial \dot{u}}\right) = 4\dot{E}^T JE\dot{u} + 4E^T J\underbrace{\dot{E}\dot{u}}_{=0} + 4E^T JE\ddot{u} = 4\dot{E}^T JE\dot{u} + 4E^T JE\ddot{u}, \tag{6}$$

the equation of motion of the shell is trivially found to be

$$\widetilde{M}\ddot{\tilde{q}} := \begin{bmatrix} mI & 0 \\ 0 & 4E^T JE \end{bmatrix}\begin{bmatrix} \ddot{w}_C \\ \ddot{u} \end{bmatrix} = \begin{bmatrix} -mge_3 \\ -8\dot{E}^T JE\dot{u} \end{bmatrix}, \tag{7}$$

where the *four*-vector $u$ at each instant of time $t$ is required (constrained) to be of unit Euclidean length. The definition of $E$ in Equation (1) shows that the matrix product $\dot{E}\dot{u}$ in Equation (6) is zero.

Equation (7) describes the unconstrained motion of the shell, $\Sigma$, under gravity. However, the components of the seven-vector $\tilde{q}$ are not independent, because the constraint $\|u(t)\| = 1$ is required on the components of the four-vector $u$ to represent the rotational motion of the shell correctly. Thus, while the use of the quaternion $u$ eliminates singularities in the description of the rotational dynamics as stated before, it adds an extra coordinate

beyond the minimum needed to describe the rotational motion of the shell. This additional coordinate makes the block diagonal mass matrix $\widetilde{M}$ singular. The four by four matrix $E^T J E$ has rank three, since

$$
\begin{aligned}
Rank(E^T J E) \quad &= Rank\left(\left(J^{1/2}E\right)^T\left(J^{1/2}E\right)\right) = Rank\left(J^{1/2}E\right) \\
&= Rank(E) = Rank\left(EE^T\right) = Rank(I_3) = 3.
\end{aligned}
\tag{8}
$$

This makes its determinant zero, and therefore the determinant of $\widetilde{M}$ also zero.

To model the shell, $\Sigma$, rolling on an arbitrarily specified rigid surface $\Gamma$ without slipping, we require it: (1) to always remain in contact with the surface and (2) to roll on the surface without slipping. These two requirements (constraints) embody the core of the problem that we are addressing. And their mathematical description becomes straightforward and simple when we use the coordinates of the geometrical center, $O$, of the shell, $\Sigma$, and the coordinates of the point of contact, $P$, between $\Sigma$ and the arbitrary surface $\Gamma$ on which it rolls. Indeed, as the shell rolls over the surface, the coordinates of both these points change with time $t$. However, as yet the coordinates of these two points do *not* appear in the seven-vector $\widetilde{q}$ that describes the configuration of our dynamical system. In order to include them as part of the configuration vector, we resort to the novel idea of adding two particles, each of zero mass, and co-locating them with the points $P$ and $O$, respectively (see Figure 2).

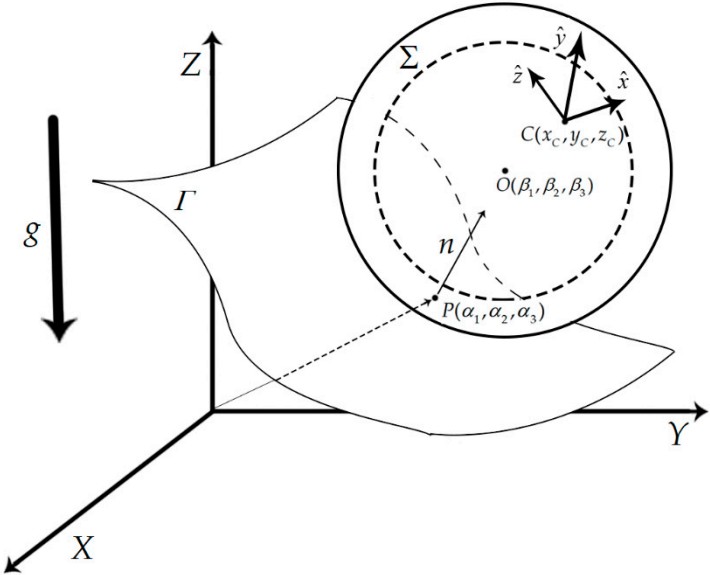

**Figure 2.** The constrained system consisting of the shell $\Sigma$ in contact with the surface $\Gamma$, a zero-mass particle $P$ located at the contact point, and a zero-mass particle $O$ located at the center of the shell. The unit normal vector perpendicular to the surface $\Gamma$ at $P$ is $n$.

Recall that zero-mass particles are never used in non-relativistic mechanics, because, from a physical viewpoint, no forces can be applied to them, and, from a mathematical viewpoint, they lead to singular mass matrices, thereby precluding the determination of the accelerations of the system. However, recent advances in analytical dynamics show that in certain situations, as we saw with Equation (7) above, we can use zero-mass particles provided the dynamical system is adequately constrained [11].

So far then, the use of quaternions in Equation (7) has added one more coordinate to the description of the dynamical system than the minimum required, along with the unit length constraint on the four-vector $u$, which is part of the system's configuration vector $\widetilde{q}$. Now, we add to this configuration vector $\widetilde{q}$ the coordinates of the points $P$ and $O$ by placing zero-mass particles at these locations, thereby obtaining a new dynamical system whose configuration vector has six more coordinates than those in $\widetilde{q}$. This makes both the

description of the constraints simpler (as the shell $\Sigma$ rolls over the surface), and it also makes the description of the equations of motion of the rolling shell on the surface much simpler to develop.

The shell rolling on the surface is thus modeled as a *multi-body system* consisting of the shell $\Sigma$ *and* two zero-mass particles, one located at the point of contact $P$ between the shell and the surface and the other located at the geometrical center $O$ of the shell.

Denoting the coordinates of a zero-mass particle located at $P$, the point of contact between the $\Sigma$ and the surface, by $\alpha = [\alpha_1, \alpha_2, \alpha_3]^T$, and the coordinates of a zero-mass particle located at the point $O$ by $\beta = [\beta_1, \beta_2, \beta_3]^T$, the configuration of this multi-body system is described by the 13-vector

$$q := \left[ w_C^T, u^T, \alpha^T, \beta^T \right]^T := \left[ \underbrace{x_c, y_c, z_c}_{coordinates\,of\,C}, \underbrace{u_0, u_1, u_2, u_3,}_{u} \underbrace{\alpha_1, \alpha_2, \alpha_3}_{coordinates\,of\,P,\,\alpha}, \underbrace{\beta_1, \beta_2, \beta_3}_{coordinates\,of\,O,\,\beta} \right]^T. \tag{9}$$

Writing the equations of motion of these two zero-mass particles that have been included is, of course, trivial. Their equations, using Newton's laws, simply state that the mass of the particle located at $P(O)$, which is zero, times its acceleration in any direction, equals zero, since no force can be applied to zero-mass particles.

The equations of motion of the multi-body system consisting of the shell $\Sigma$ and the two zero-mass particles (located at $O$ and $P$) are therefore obtained directly as

$$M\ddot{q} := \begin{bmatrix} mI & 0 & 0 & 0 \\ 0 & 4E^T JE & 0 & 0 \\ 0 & 0 & M_P & 0 \\ 0 & 0 & 0 & M_O \end{bmatrix} \begin{bmatrix} \ddot{w}_C \\ \ddot{u} \\ \ddot{\alpha} \\ \ddot{\beta} \end{bmatrix} \begin{bmatrix} -mge_3 \\ -8\dot{E}^T JE\dot{u} \\ 0 \\ 0 \end{bmatrix} := Q(q, \dot{q}). \tag{10}$$

Notice that the equations of motion comprise those already obtained in Equation (7) along with the trivial equations

$$M_P \ddot{\alpha} = 0, \text{ and } M_O \ddot{\beta} = 0, \tag{11}$$

for the two zero-mass particles, where the three by three (diagonal) matrices $M_P = M_O = 0$. The right-hand sides of each equation in Equation (11) is the zero three-vector, since, as mentioned before, no force can be applied to a zero-mass particle. It should be noted that the 13 by 13 mass matrix $M$ is singular. So far, we have one constraint that the four-vector $u$ must satisfy, namely $\|u(t)\| = 1$.

We assume that the surface $\Gamma$ is smooth enough that at every point on it all the second order partial derivatives of $\Gamma(X, Y, Z) = 0$ exist. In addition, we assume that the shell touches the surface at just a single point. In other words, when the shell $\Sigma$ with center $O$ at $\beta(t) = [\beta_1(t), \beta_2(t), \beta_3(t)]^T$ touches the surface $\Gamma$ during its motion at a point $P$ with coordinate $\alpha(t) = [\alpha_1(t), \alpha_2(t), \alpha_3(t)]^T$ so that $\Gamma(\alpha_1, \alpha_2, \alpha_3) = 0$, it touches *no* other point of the surface $\Gamma$. Thus, we assume that the relation between the two points $P$ and $O$, whose coordinates are $\alpha(t)$ and $\beta(t)$, respectively, is given by

$$\beta = \alpha + rn(\alpha) := f(\alpha), \tag{12}$$

where $n$ is the unit normal to the surface $\Gamma$ at the point of contact $\alpha$. Furthermore, this equation provides a unique value for $\beta$ (point $O$) given a value of $\alpha$ (point $P$) and the function f is invertible. Therefore, the Jacobian of f is nonsingular at every point on the surface $\Gamma$.

The normal to the surface $\Gamma$ at a point of contact $\alpha$ is given by the three-vector

$$k(\alpha) = \left[\begin{array}{ccc} \frac{\partial \Gamma}{\partial X} & \frac{\partial \Gamma}{\partial Y} & \frac{\partial \Gamma}{\partial Z} \end{array}\right]^T \Bigg|_{\left[\begin{array}{c} X \\ Y \\ Z \end{array}\right] = \left[\begin{array}{c} \alpha_1 \\ \alpha_2 \\ \alpha_3 \end{array}\right]} \tag{13}$$

where $\Gamma(X, Y, Z) = 0$ is expressed (arranged) such that the Z-component of the three-vector $k$ is taken to be positive. At the point of contact, the unit normal vector $n$ to the surface is then obtained as

$$n(\alpha) = \frac{k}{\sqrt{k^T k}} = \frac{k}{\|k\|}. \tag{14}$$

Appendix A explicitly gives the Jacobian of the function $f(\alpha)$ defined in Equation (12), which we denote by $J_f$. As stated before, $J_f^{-1}$ exists at every point $\alpha$ on the surface $\Gamma$.

### 2.2. Description of the Constraints

In this subsection, we describe the constraints that the components of the configuration vector must satisfy. They can be categorized as those coming from:

(1)   The use of quaternions to describe the rotational dynamics of the shell;
(2)   The description of the location of the two zero-mass particles, one placed at the point of contact $P$ between the shell $\Sigma$ and the surface $\Gamma$ and the other at the center, $O$, of $\Sigma$;
(3)   The constraints relevant to the physical conditions that must be satisfied by the shell to roll without slipping on the surface;
(4)   Additional constraints that might be redundant but are consistent with all the other existing constraints, and/or constraints that may be added to the system to, for example, further physically constrain the motion of the shell $\Sigma$.

We consider each of these categories in turn.

(1)   Quaternion Constraint

The constraint on the four-vector $u$, as mentioned before, is described by

$$\varphi_1 := u_0(t)^2 + u_1(t)^2 + u_2(t)^2 + u_3(t)^2 - 1 = 0. \tag{15}$$

A suitable form of constraint can be obtained by taking the second time derivative of Equation (15) to yield

$$A_1 \ddot{q} := \left[\begin{array}{cccc} 0_{1\times 3} & | & u^T & | & 0_{1\times 3} & | & 0_{1\times 3} \end{array}\right] \ddot{q} = -\dot{u}^T \dot{u} := b_1. \tag{16}$$

We call $\varphi_1(t)$ the 'Quaternion Constraint'.

(2)   Location of the two zero-mass particles at $P$ and $O$

    (i)       Location of the zero-mass particle at $P$

The first zero-mass particle is co-located at the point $P$ that lies at the point of contact between the shell and the surface, and therefore its coordinate $\alpha$ must satisfy the equation of the surface. This leads to the constraint

$$\varphi_2 := \Gamma(\alpha_1, \alpha_2, \alpha_3) = 0, \tag{17}$$

which we shall refer to as the 'Surface Constraint'.

As before, the second time derivative of Equation (17) in term of $k$, defined in Equation (13), is given by

$$A_2 \ddot{q} := \left[\begin{array}{cccc} 0_{1\times 3} & | & 0_{1\times 3} & | & k^T & | & 0_{1\times 3} \end{array}\right] \ddot{q} = -\dot{k}^T \dot{\alpha} = -\dot{\alpha}^T \left[\frac{\partial k}{\partial \alpha}\right] \dot{\alpha} := b_2, \tag{18}$$

noting that the matrix $\frac{\partial k}{\partial \alpha}$ is symmetric.

(ii)　　Location of the zero-mass particle at $O$

For the second zero-mass particle to be co-located with the point $O$, which is the geometric center of the shell $\Sigma$ (see Figure 2), the distance $OP$ must be $r$ and $O$ must lie along the normal to the surface at $P$. Hence, we obtain the relation

$$\varphi_3 := \beta - \alpha - rn = 0. \tag{19}$$

We call this the 'Tangency Constraint'. Again, taking the second derivative of Equation (19) with respect to time $t$, we can write

$$A_3 \ddot{q} = b_3. \tag{20}$$

The matrix $A_3$ and the three by one column vector $b_3$ are obtained in Appendix A as

$$A_3 = \begin{bmatrix} 0_{3\times3} & | & 0_{3\times4} & | & J_f & | & -I_{3\times3} \end{bmatrix}, \text{ and } b_3 = -r\left[ \Delta\frac{d}{dt}\left(\frac{\partial k}{\partial \alpha}\right)\dot{\alpha} + k\delta - \frac{2\dot{k}^T k}{(k^T k)^{3/2}}\frac{\partial k}{\partial \alpha}\dot{\alpha}\right] \tag{21}$$

where the Jacobian $J_f = \left[ I_{3\times3} + r\Delta\frac{\partial k}{\partial \alpha}\right]$, and the expressions for $\Delta$ and $\delta$ are given in Appendix A.

(3)　　Physical Constraints

(i)　　Constraint on $w_C$

As seen from Equation (10), the coordinate $w_c$ of the center of mass $C$ of the shell in the unconstrained system is uncoupled from the coordinates $u, \alpha, \beta$. However, when the shell $\Sigma$ rolls over the surface $\Gamma$, $w_c$ depends on the (rotational) orientation of the shell and the location of the point of contact (or alternately as seen from Equation (12), the location of the point $O$). The zero-mass particle co-located at $O$ simplifies this relation and we have

$$\varphi_4 := \beta + S\rho - w_C = 0 \tag{22}$$

where $S$ is the active rotation matrix, which can be written in terms of the quaternion components as

$$S = \begin{bmatrix} u_0^2 + u_1^2 - u_2^2 - u_3^2 & 2(u_1 u_2 - u_0 u_3) & 2(u_1 u_3 + u_0 u_2) \\ 2(u_1 u_2 + u_0 u_3) & u_0^2 - u_1^2 + u_2^2 - u_3^2 & 2(u_2 u_3 - u_0 u_1) \\ 2(u_1 u_3 - u_0 u_2) & 2(u_2 u_3 + u_0 u_1) & u_0^2 - u_1^2 - u_2^2 + u_3^2 \end{bmatrix} \tag{23}$$

and $\rho = [\rho_1, \rho_2, \rho_3]^T$ is the vector that starts from the center of the shell $O$ and points to the center of mass $C$ of the shell; its components are measured in the body-fixed coordinate frame $\hat{x}\hat{y}\hat{z}$ and are therefore constant and they depend on the distribution of the mass of in the shell $\Sigma$ (see Figure 1). We call the constraint, $\varphi_4$, the 'Geometric Center Constraint'. We write the constraint Equation (22) in suitable form by taking its second derivative with respect to time. Defining $\dot{S}\rho = C_1 u$ where

$$C_1 = 2\begin{bmatrix} u_0\rho_1 - u_3\rho_2 + u_2\rho_3 & u_1\rho_1 + u_2\rho_2 + u_3\rho_3 & -u_2\rho_1 + u_1\rho_2 + u_0\rho_3 & -u_3\rho_1 - u_0\rho_2 + u_1\rho_3 \\ u_3\rho_1 + u_0\rho_2 - u_1\rho_3 & u_2\rho_1 - u_1\rho_2 - u_0\rho_3 & u_1\rho_1 + u_2\rho_2 + u_3\rho_3 & u_0\rho_1 - u_3\rho_2 + u_2\rho_3 \\ -u_2\rho_1 + u_1\rho_2 + u_0\rho_3 & u_3\rho_1 + u_0\rho_2 - u_1\rho_3 & -u_0\rho_1 + u_3\rho_2 - u_2\rho_3 & u_1\rho_1 + u_2\rho_2 + u_3\rho_3 \end{bmatrix} \tag{24}$$

and $C_2 = \dot{C}_1$ the second derivative of Equation (22) can be written as

$$\ddot{\beta} + C_1 \ddot{u} + C_2 \dot{u} - \ddot{w}_C = 0 \tag{25}$$

which can be recast in the form

$$A_4 \ddot{q} := \begin{bmatrix} -I_{3\times3} & | & C_1 & | & 0_{3\times3} & | & I_{3\times3} \end{bmatrix} \ddot{q} = -C_2 \dot{u} := b_4. \tag{26}$$

(ii)    The Rolling No-Slip Constraint

The shell $\Sigma$ rolling on the surface $\Gamma$ without slipping requires the non-holonomic constraint

$$\varphi_5 := \dot{\beta} + r\tilde{n}S\omega = 0 \tag{27}$$

to be satisfied at each instant of time $t$. We note that the normal to the surface is $n(\alpha(t))$ and it is therefore a function of the time, $t$, as the shell rolls over the surface. The active rotation matrix $S$ is given in Equation (23). The matrix $\tilde{n}(t)$ is the three by three skew-symmetric matrix obtained from the components of the unit three-vector $n$ in the XYZ coordinate frame in Equation (14); it is given by

$$\tilde{n} = \begin{bmatrix} 0 & -n_Z & n_Y \\ n_Z & 0 & -n_X \\ -n_y & n_X & 0 \end{bmatrix} \tag{28}$$

This notation of a tilde above a three-vector to denote the skew-symmetric matrix of its components shown on the right hand side of (28) will be used throughout this paper. Equation (27) states that the instantaneous velocity of the point on the shell that touches the surface $\Gamma$ is zero. The second term in Equation (27) is the relative velocity of this point while the shell is rotating with angular velocity $\omega$. The three-vector $\omega$ contains the components of the angular velocity in the body-fixed $\hat{x}\hat{y}\hat{z}$ coordinate frame. Differentiating Equation (27) twice with respect to time $t$, we obtain

$$\ddot{\beta} + 2r\dot{\tilde{n}}SE\dot{u} + 2r\tilde{n}\dot{S}E\dot{u} + 2r\tilde{n}S\underbrace{\dot{E}\dot{u}}_{=0} + 2r\tilde{n}SE\ddot{u} = 0, \tag{29}$$

which, upon noting that $SS^T = I$, can be rewritten as

$$\ddot{\beta} + 2r\dot{\tilde{n}}SE\dot{u} + 2r\tilde{n}S\underbrace{S^T\dot{S}}_{=\tilde{\omega}}E\dot{u} + 2r\tilde{n}SE\ddot{u} = 0. \tag{30}$$

Since $S^T\dot{S} = \tilde{\omega}$, $\omega = 2E\dot{u}$, and $\tilde{\omega}\omega = 0$, the third term on the left in Equation (30) is zero and the equation can be simplified to

$$A_5\ddot{q} := \begin{bmatrix} 0_{3\times3} & | & 2r\tilde{n}SE & | & 0_{3\times3} & | & I_{3\times3} \end{bmatrix}\ddot{q} = -2r\dot{\tilde{n}}SE\dot{u} := b_5. \tag{31}$$

Constraints (16), (20), (26), (18), and (31), obtained so far, can be expressed as a system of equations in form of

$$A\ddot{q} := \begin{bmatrix} A_1 \\ A_2 \\ A_3 \\ A_4 \\ A_5 \end{bmatrix}\ddot{q} = \begin{bmatrix} b_1 \\ b_2 \\ b_3 \\ b_4 \\ b_5 \end{bmatrix} := b, \tag{32}$$

where the matrix $A$ is an 11 by 13 matrix and vector $b$ is an 11-vector (11 by 1). These five sets of constraint equations are sufficient to model the spherical shell $\Sigma$ rolling on the surface $\Gamma$ without slipping. It should be noted that the rows of matrix $A$ do not have to be independent and multiple consistent constraints can be imposed on the system.

(4)    Additional Constraints

(i)    Constraints Related to Known Conserved Quantities

One of the significant advantages of this methodology is that even additional constraints that are not independent from of the existing constraints can be added to the system. In other words, the rows of matrix $A$ need not be independent. This capability lets us make the numerical model more consistent with conserved quantities that are known to exist

during the evolving motion of the system. For instance, in the modeling of a system whose energy is conserved, the energy conservation equation can be added to the rows of matrix $A$ as an additional constraint. Since there is no dissipation or injection of energy to our system, energy is conserved. Using Equations (2) and (3), the equation that states that the total amount of energy, $E(t)$, of the system at each instant of time $t$ remains constant can be written as

$$\phi_E := E(t) - E(0) = \frac{m}{2}\dot{w}_C{}^T\dot{w}_C + 2\dot{u}^T E^T JE\dot{u} + mgw_C^T e_3 - E(0) = 0, \tag{33}$$

and taking the time derivative of Equation (33) gives the relation

$$A_E\ddot{q} = b_E \tag{34}$$

where

$$A_E = \left[ m\ddot{w}_C^T \mid 4\dot{u}^T E^T JE \mid 0_{1\times 3} \mid 0_{1\times 3} \right] \text{ and } b_E = mg\dot{w}_C^T e_3. \tag{35}$$

Hence, Equation (34) can be added to the set of constraints given in Equation (32), so that the augmented set of constraints is given by the system of equations

$$A\ddot{q} := \begin{bmatrix} A_1 \\ A_2 \\ A_3 \\ A_4 \\ A_5 \\ A_E \end{bmatrix} \ddot{q} = \begin{bmatrix} b_1 \\ b_2 \\ b_3 \\ b_4 \\ b_5 \\ b_E \end{bmatrix} := b, \tag{36}$$

where the matrix $A$ is now 12 by 13 and the vector $b$ is 12 by 1.

(ii)     No-Spin Constraint

Although the constraints in Equation (32) are sufficient to model the motion of the shell rolling on a prescribed surface without slipping, depending on the situation at hand, the motion of the shell can be further restricted by imposing additional constraints. For instance, the spin of the shell $\Sigma$ about the normal vector $n$ to the surface $\Gamma$ can be prevented during its rolling motion by the inclusion of an additional constraint. As shown in Appendix B, the components of the angular velocity of the shell $\Sigma$ at each instant of time in the $\hat{x}\hat{y}\hat{z}$ body-frame and in the XYZ inertial frame can be expressed as

$$\omega = S^T\left(\tilde{n}\frac{\dot{\beta}}{r} + \lambda n\right) \tag{37}$$

and

$$\overline{\omega} = S\omega = \left(\tilde{n}\frac{\dot{\beta}}{r} + \lambda n\right), \tag{38}$$

respectively. The two three-vectors on the right hand side in the last equality in Equation (38) are orthogonal to each other since $n^T\tilde{n} = (\tilde{n}^T n)^T = -(\tilde{n}n)^T = 0$. The component of $\overline{\omega}$ along the unit normal $n$ to the surface $\Gamma$ is $n^T\overline{\omega}$. Thus, from Equation (38) we see that the component of the angular velocity $\overline{\omega}$ (in the inertial XYZ frame) normal to the surface $\Gamma$ is $\lambda$. In addition, the tangential component of the angular velocity $\overline{\omega}$ is determined in terms of the velocity three-vector $\dot{\beta}$ of the center $O$ of the shell. When $\lambda(t) \equiv 0$, the angular velocity of the shell is thus seen to depend only on the velocity of its center $O$. We refer to the component $\lambda = n^T\overline{\omega} = n^T S\omega$ of the angular velocity $\overline{\omega}$ about an axis normal to the surface $\Gamma$ as the '*spin velocity*' of the shell $\Sigma$ throughout this paper. Noting from Equation (14) that the three-vector $n = k/\|k\|$, we have $\lambda = n^T S\omega = k^T S\omega/\|k\|$. Thus, to constrain the shell

from 'spinning' about the normal to the surface $\Gamma$ when it rolls, we use the non-holonomic constraint [24]

$$\varphi_{NS} := k^T S \omega = 0. \tag{39}$$

The subscript '*NS*' signifies the 'No-Spin' constraint. This constraint simply states that the component of the angular velocity vector along the vector *n* is zero during the motion of the shell. Differentiating Equation (39) twice with respect to time, we obtain

$$\dot{k}^T S E \dot{u} + k^T \underbrace{\dot{S} E \dot{u}}_{=0} + k^T S \underbrace{\dot{E} \dot{u}}_{=0} + k^T S E \ddot{u} = 0. \tag{40}$$

The first bracketed term on the left hand side computes to $\dot{S} E \dot{u} = S(S^T \dot{S})(E \dot{u}) = S \widetilde{\omega} \omega / 2 = 0$, where we have used Equation (1) in the second-last equality. Equation (40) can be rearranged as

$$A_{NS} \ddot{q} := \begin{bmatrix} 0_{1\times 3} & | & k^T S E & | & 0_{1\times 3} & | & 0_{1\times 3} \end{bmatrix} \ddot{q} = - \dot{\alpha}^T \left[ \frac{\partial k}{\partial \alpha} \right] S E \dot{u} := b_{NS}. \tag{41}$$

If this additional No-Spin constraint is required to be imposed, one simply includes Equation (41) in the set of constraints given earlier in Equation (36). This gives the new set of constraint equations

$$A \ddot{q} := \begin{bmatrix} A_1 \\ A_2 \\ A_3 \\ A_4 \\ A_5 \\ A_E \\ A_{NS} \end{bmatrix} \ddot{q} = \begin{bmatrix} b_1 \\ b_2 \\ b_3 \\ b_4 \\ b_5 \\ b_E \\ b_{NS} \end{bmatrix} := b \tag{42}$$

where the matrix $A$, upon this inclusion, is now a 13 by 13 matrix and the column vector $b$ is a 13-vector. It should be noted that in the presence of the No-Spin constraint, the energy of the constrained system is still conserved, because at each instant of time the spin about the normal $n$ to the surface $\Gamma$ is zero and therefore there is no work done by the constraint torque about the normal; Equations (34) and (33) therefore continue to be applicable to the dynamical system. As seen above, the methodology developed here allows the easy handling of additional constraints. In a simple and straightforward manner, it permits one to determine the effect of the addition or exclusion of one or more constraints on the evolutionary dynamics of the system.

*2.3. Description of the Constrained Multi-Body System*

Given the matrices $M$, $Q$, $A$ and $b$ that are derived based on the description of the unconstrained system and the constraint equations that the unconstrained system is required to satisfy, we next determine the explicit equations of motion of the constrained dynamical system.

Though the mass matrix $M$ of the unconstrained equations of motion is singular, as stated before, the equations of motion of the unconstrained system in the presence of the constraints can provide the acceleration of the constrained system uniquely. This (unique) acceleration $\ddot{q}$ that satisfies d'Alembert's principle is explicitly given by [15]:

$$\ddot{q} = \begin{bmatrix} (I - A^+ A)M \\ A \end{bmatrix}^+ \begin{bmatrix} Q \\ b \end{bmatrix}, \tag{43}$$

where $X^+$ denotes the Moore–Penrose (MP) inverse of the matrix $X$ [12,32]. However, Equation (43) is valid *if and only if* the $m$ by $n$ matrix $\overline{M}$ defined as

$$\overline{M} := \begin{bmatrix} M \\ A \end{bmatrix} \tag{44}$$

is such that

$$Rank(\overline{M}) = n \tag{45}$$

where $n$ is the dimension of the vector $q$ in Equation (9).

It should be noted that the row dimension of the matrix $\overline{M}$ in Equation (44) depends on the row dimension of $A$—the number and nature of constraints included in the modeling—and the dimension of the square matrix $M$; its column dimension is that of the column vector, $q$, which in our case is 13. In the following Result, we next show that Equation (45) *is* satisfied and that $Rank(\overline{M}) = n = 13$, thereby providing an analytical proof that Equation (43) is the closed form equation that describes the motion of a shell moving on an arbitrarily prescribed surface.

**Remark 1.** *Besides the approach used in this paper and given in Equation (43), there is an alternative way of modeling the dynamical behavior of systems whose unconstrained equations of motion have singular mass matrices (see, for example, Refs. [24,29,30]).*

**Result 1.** *With the matrices M and A given in Equations (10) and (36) or (42), respectively, the matrix $\overline{M}$ has full column rank.*

**Proof.** The matrix $\overline{M}$, in which the matrix $A$ is either given by Equation (36) or Equation (42), has full column rank if the truncated matrix, $\hat{M}$, obtained from $\overline{M}$ by discarding some of its rows

$$\hat{M} = \begin{bmatrix} M \\ A_1 \\ A_3 \\ A_4 \end{bmatrix}_{20 \times 13} = \begin{bmatrix} M_{13 \times 13} \\ 0_{1 \times 3} \mid u^T_{1 \times 4} \mid 0_{1 \times 3} \mid 0_{1 \times 3} \\ 0_{3 \times 3} \mid 0_{3 \times 4} \mid J_f \mid -I_{3 \times 3} \\ -I_{3 \times 3} \mid C_1 \mid 0_{3 \times 3} \mid I_{3 \times 3} \end{bmatrix} \tag{46}$$

has full column rank. The three by three matrix $J_f = [I + r\Delta\frac{\partial k}{\partial \alpha}]$ in Equation (46) is the non-singular Jacobian matrix (see Equation (21) and Appendix A) of $f(\alpha)$ that is defined in Equation (22). Our aim is to show that $\hat{M}$ has full column rank, i.e., $Rank(\hat{M}) = $ column-size of $q = 13$. □

We begin by showing that there exists a matrix $D$ that is compatible with $\hat{M}$ such that the square matrix $D\hat{M}$ has rank = 13. Consider the 13 by 20 matrix $D$ given by

$$D = [D_M \mid D_1 \mid D_3 \mid D_4] \tag{47}$$

where

$$D_M = \begin{bmatrix} \frac{1}{m} I_{3 \times 3} & 0 & 0 & 0 \\ 0 & \frac{1}{4} E^T J^{-1} E_{4 \times 4} & 0 & 0 \\ 0 & 0 & 0_{3 \times 3} & 0 \\ 0 & 0 & 0 & 0_{3 \times 3} \end{bmatrix}_{13 \times 13} \tag{48}$$

and

$$D_1 = \begin{bmatrix} 0_{3 \times 1} \\ u_{4 \times 1} \\ 0_{3 \times 1} \\ 0_{3 \times 1} \end{bmatrix}_{13 \times 1}, \quad D_3 = \begin{bmatrix} 0_{3 \times 3} \\ 0_{4 \times 3} \\ J_f^{-1} \\ 0_{3 \times 3} \end{bmatrix}_{13 \times 3}, \quad D_4 = \begin{bmatrix} 0_{3 \times 3} \\ 0_{4 \times 3} \\ J_f^{-1} \\ I_{3 \times 3} \end{bmatrix}_{13 \times 3}. \tag{49}$$

Noting that $EE^T = I_3$, $E^T E + uu^T = I_4$, the product $D\hat{M}$ can be written as

$$
\begin{aligned}
D\hat{M} \quad &= D_M M + D_1 A_1 + D_3 A_3 + D_4 A_4 \\
&= \begin{bmatrix} I & 0 & 0 & 0 \\ 0 & E^T E & 0 & 0 \\ 0 & 0 & 0 & 0 \\ 0 & 0 & 0 & 0 \end{bmatrix} + \begin{bmatrix} 0 & 0 & 0 & 0 \\ 0 & uu^T & 0 & 0 \\ 0 & 0 & 0 & 0 \\ 0 & 0 & 0 & 0 \end{bmatrix} + \begin{bmatrix} 0 & 0 & 0 & 0 \\ 0 & 0 & 0 & 0 \\ 0 & 0 & I & -J_f^{-1} \\ 0 & 0 & 0 & 0 \end{bmatrix} + \begin{bmatrix} 0 & 0 & 0 & 0 \\ 0 & 0 & 0 & 0 \\ -J_f^{-1} & J_f^{-1} C_1 & 0 & J_f^{-1} \\ -I & C_1 & 0 & I \end{bmatrix} \\
&= \begin{bmatrix} I_3 & 0 & 0 & 0 \\ 0 & I_4 & 0 & 0 \\ -J_f^{-1} & J_f^{-1} C_1 & I_3 & 0 \\ -I & C_1 & 0 & I_3 \end{bmatrix}_{13 \times 13}
\end{aligned} \tag{50}
$$

Since the square matrix $D\hat{M}$ is a lower-triangular matrix with all its diagonal elements equal to one, its determinant is one, and so nonsingular, hence $Rank(D\hat{M}) = 13$. However, $Rank(\hat{M}) \geq Rank(D\hat{M}) = 13$. In addition, $\hat{M}$ is a 20 by 13 matrix, therefore $Rank(\hat{M}) \leq 13$. Since both these last two inequalities must hold, $Rank(\hat{M}) = 13$ and $\hat{M}$ has full column rank.

This result proves that the (unique) acceleration $\ddot{q}$ of the constrained system is given by Equation (43).

### 2.4. Explicit Equations of Motion

We have thus analytically proved that using the matrices $M$ and $A$, and the column vectors $Q$ and $b$, which are given explicitly in Equations (10) and (36) (or Equation (42)), Equation (43) is the *closed form equation* that describes the motion of a shell $\Sigma$ as it rolls without slipping on an arbitrarily prescribed surface $\Gamma$. This equation is of course valid as long as the shell does not leave the surface during its traverse over it, as required by the constraint in Equation (17).

### 2.5. Determination of the Generalized Constraint Forces and the Generalized Reaction Provided by the Surface $\Gamma$

The methodology used here, also, provides an explicit equation for the *constraint force* $Q^C$, which is the generalized force 13-vector applied to the system to satisfy the constraints. It is simply obtained as [15]

$$
Q^C := \left[ \left[ Q_{w_C}^C \right]^T \quad \left[ Q_u^C \right]^T \quad \left[ Q_\alpha^C \right]^T \quad \left[ Q_\beta^C \right]^T \right]^T = M\ddot{q} - Q. \tag{51}
$$

Equation (51) shows the components of column vector $Q^C$ that correspond to the respective generalized coordinates $w_C$, $u$, $\alpha$, and $\beta$.

The three-vector $Q_{w_C}^C(t)$, which corresponds to the coordinate of the center of mass of the shell $w_C$, gives the components (in the inertial XYZ frame) of the constraint force applied at the center of mass $C$; the four-vector $Q_u^C(t)$, which corresponds to the coordinate $u$, gives the components of the 'quaternion constraint torque' acting at $C$. To obtain the components of the physical constraint torque $T^C$ (in the body-fixed $\hat{x}\hat{y}\hat{z}$ frame) from $Q_u^C(t)$, we use the relation (see Ref. [16])

$$
T^C = \left[ T_{\hat{x}}^C \, T_{\hat{y}}^C \, T_{\hat{z}}^C \right]^T = \frac{1}{2} E Q_u^C. \tag{52}
$$

Thus, we obtain the components of the constraint force, $Q_{w_C}^C(t)$ and the components of the constraint torque, $T^C(t)$, acting at the point $C$ that are required in order for the shell to move over the surface while satisfying the imposed constraints.

It should be noted that the components of the three-vector $Q_{w_C}^C(t)$ in Equation (52) are expressed in the inertial XYZ coordinate frame, while the components of the three-vector $T^C$ in Equation (52) are expressed in the body-fixed $\hat{x}\hat{y}\hat{z}$ coordinate frame.

We now determine from the constraint force vector, $\overrightarrow{Q_{w_C}^C}$, and the constraint torque vector, $\overrightarrow{T^C}$, the reaction force and torque provided by the surface $\Gamma$ on the shell $\Sigma$. This

is easily performed by simply moving the (generalized) force system $[\vec{Q}^C_{w_C}, \vec{T}^C]$ acting at point $C$ to an equivalent force system $[\vec{Q}^P, \vec{T}^P]$ acting at the point of contact $P$.

This yields the reaction force $\vec{Q}^P = \vec{Q}^C_{w_C}$. The component of the force vector $\vec{Q}^P$ along the normal to the surface $\Gamma$ (in the XYZ frame) is therefore the three-vector $(n^T Q^C_{w_C})n$; its magnitude is $n^T Q^C_{w_C}$. The component of $\vec{Q}_P$ tangent to $\Gamma$, which is the frictional force exerted by $\Gamma$ so that the shell rolls without slipping, is similarly given by the three-vector $(I_3 - nn^T)Q^C_{w_C}$ (in the XYZ frame) whose magnitude is $\|(I_3 - nn^T)Q^C_{w_C}\|$.

Additionally, the equivalent torque vector at $P$ is given by

$$\vec{T}^P = \vec{T}^C + (\vec{\rho} + r\,\vec{n}) \times \vec{Q}^C_{w_C}. \tag{53}$$

The components of $\vec{T}^P$ in the $\hat{x}\hat{y}\hat{z}$ body-frame can therefore be written as

$$T^P_{\hat{x}\hat{y}\hat{z}} := \begin{bmatrix} T^P_{\hat{x}} \\ T^P_{\hat{y}} \\ T^P_{\hat{z}} \end{bmatrix} = \begin{bmatrix} T^C_{\hat{x}} \\ T^C_{\hat{y}} \\ T^C_{\hat{z}} \end{bmatrix} + \widetilde{\rho} S^T(Q^C_{w_C}) + rS^T(\widetilde{n}Q^C_{w_C}). \tag{54}$$

The three-vectors in brackets on the right hand side of Equation (54) are components of the vectors $\vec{Q}^C_{w_C}$ and $\vec{n} \times \vec{Q}^C_{w_c}$ in the XYZ coordinate frame; they are transformed to the corresponding components in the $\hat{x}\hat{y}\hat{z}$ coordinate frame through pre-multiplication by $S^T$, as shown.

Similarly, the components of $\vec{T}^P$ in the XYZ inertial frame can be obtained from Equation (53) by first expressing the vectors $\vec{T}^C$ and $\vec{\rho}$ in the XYZ frame to yield

$$T^P_{XYZ} := \begin{bmatrix} T^P_X \\ T^P_Y \\ T^P_Z \end{bmatrix} = ST^C + [S\rho + r\widetilde{n}]Q^C_{w_c}. \tag{55}$$

Here, the three-vectors $T^C$ and $\rho$ are the components of $\vec{T}^C$ and $\vec{\rho}$, respectively, in the $\hat{x}\hat{y}\hat{z}$ frame and they are expressed in the XYZ frame through pre-multiplication by $S$. Note that the tilde in Equation (55) is over the three-vector $S\rho$.

Alternatively, since $T^P_{XYZ} = ST^P_{\hat{x}\hat{y}\hat{z}}$ and $SS^T = I$, from Equation (54) the components of the torque at the point $P$ in the XYZ frame are directly given by

$$T^P_{XYZ} = ST^C + \left[S\widetilde{\rho}S^T + r\widetilde{n}\right]Q^C_{w_C}. \tag{56}$$

We have shown that the expressions given on the right hand sides of Equations (55) and (56) for $T^P_{XYZ}$ must be equivalent. This can also be shown directly by interpreting $\widetilde{\rho}$ as the matrix representation of a linear operator expressed in the basis set of the body-fixed $\hat{x}\hat{y}\hat{z}$ coordinate system. When expressed in the basis set of the XYZ inertial coordinate system, the matrix representation of this operator is therefore $S\widetilde{\rho}S^T$, as shown in Equation (56). As with the reaction force vector, $\vec{Q}_P$, the magnitude of the component (in the XYZ frame) of the torque vector $\vec{T}^P$ in the normal direction, $n$, to the surface $\Gamma$ is $(n^T T^P_{XYZ})$, and the magnitude of the component tangent to the surface (in the XYZ frame) is $\|(I_3 - nn^T)T^P_{XYZ}\|$.

In this multi-body system, since each of the particles located at $\alpha$ and $\beta$ have zero mass, their corresponding constraint forces $Q^C_\alpha$ and $Q^C_\beta$ are expected to each be zero three-vectors; this is corroborated by the computational results obtained below.

## 3. Computational Results

Three numerical examples are considered in this section. They investigate the motion of a non-uniform rigid spherical shell rolling under gravity without slipping on a dimpled bowl-shaped surface. The equations of motion in all the numerical examples are obtained from Equation (43).

All the computations are performed on the MATLAB platform using a modified variable-step (4, 5) Runge–Kutta integrator with relative and absolute error tolerances of $10^{-12}$ and $10^{-15}$, respectively. The gravitational acceleration, g, is taken to be 9.81 m/s$^2$ in the simulations.

*Description of the Shell* $\Sigma$: We describe here the properties of the non-uniform rigid shell $\Sigma$ considered in all the simulations in this section. The shell has an outer radius of $r = 9$ cm and an inner radius of $r_i = 8$ cm. One half of the shell $\Sigma$ is made of silver with a density of $d_H = 10{,}490$ kg/m$^3$ and the other half is made of iron with a density of $d_L = 7874$ kg/m$^3$. The origin of the body-fixed frame $\hat{x}\hat{y}\hat{z}$ is placed at the shell's center of mass C, and the $\hat{x}$-axis of the body-fixed frame lies along the line of symmetry with its positive direction pointing in the direction of the lower density hemisphere of the shell. The center O of the shell is located along the $\hat{x}$-axis and it is approximately at a distance of 0.61 cm from C.

*Description of Surface* $\Gamma$: An asymmetric multi-dimpled bowl-shaped surface is considered in this study. It is described by $\Gamma(X, Y, Z) = 0$, where

$$\Gamma(X, Y, Z) = Z - \frac{h_1 \sin(h_2 \pi X)}{1 + X^2} - \frac{h_3 \sin(h_4 \pi Y)}{1 + Y^2} - h_5(X^2 + Y^2), \qquad (57)$$

with $h_1 = 0.04$, $h_2 = 1.2\sqrt{2}$, $h_3 = 0.05$, $h_4 = 1.6$ and $h_5 = 0.03$.

Figure 3a shows a three-dimensional plot of the surface $\Gamma$. The vertical arrow marks the location of the initial point of contact of the shell with the surface for all the numerical simulations that follow. Figure 3b shows a contour plot of the surface illustrating its asymmetry.

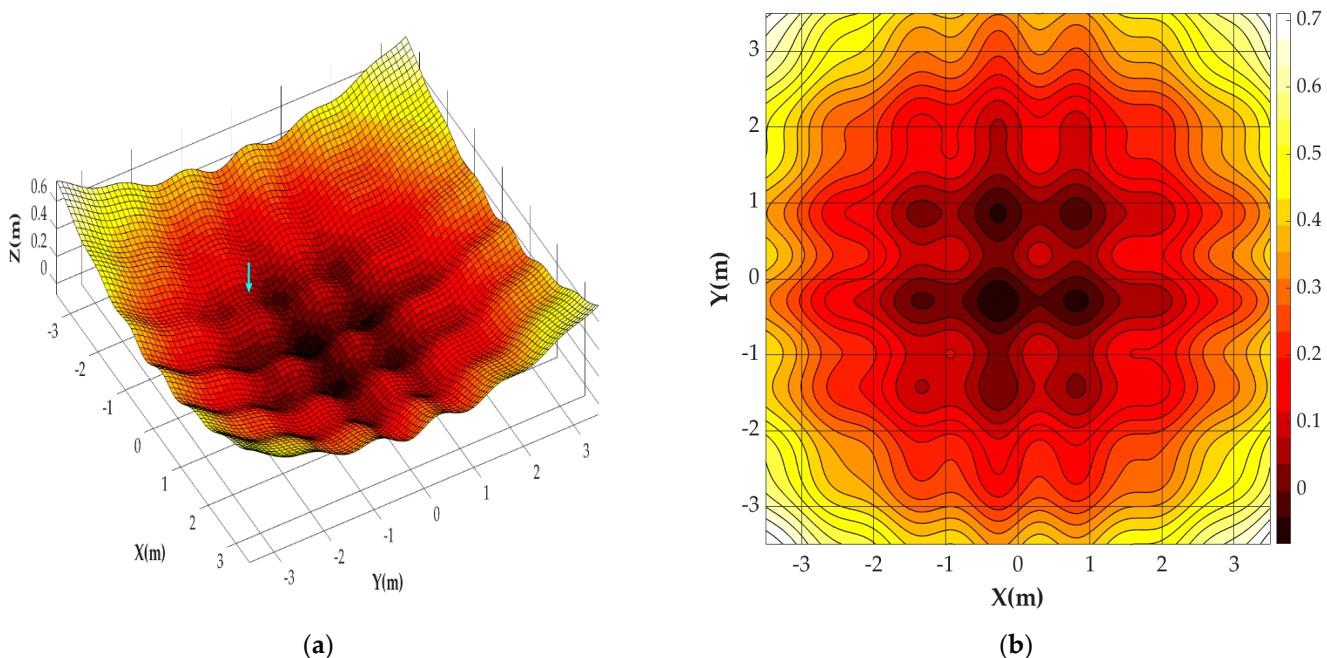

(a)  (b)

**Figure 3.** (**a**) 3D plot of the asymmetrical multi-dimpled bowl-shaped surface $\Gamma(X, Y, Z) = 0$. The vertical arrow shows the initial point of contact of the shell with the surface for all the simulations. (**b**) Contour plot of the surface.

### 3.1. Three Examples and Four Initial Shell Orientations

Three different examples are considered in this study and the motion of the shell $\Sigma$ over the surface $\Gamma$ given in Equation (57) is simulated for a duration of 25 s. We now describe these examples with special emphasis on how they differ from one another.

At the initial time $t = 0$, the point $P$ at which the shell touches the surface has components given by $\alpha_1(0) = -1.5$ m and $\alpha_2(0) = -0.75$ m in the inertial XYZ coordinate frame. The initial velocity components of the (geometric) center $O$ of the shell (also in the XYZ frame) are $\dot{\beta}_1 = 1.5$ m/s and $\dot{\beta}_2 = 1$ m/s. These initial conditions are used in *all* the examples that follow. The initial location of the point of contact $P$ of the shell with the surface $\Gamma$ for all the simulations is shown by the vertical arrow in Figure 3a. The rest of the initial conditions are determined by using the constraint equations.

**Example 1.** *At the start of the motion ($t = 0$), the shell is given no spin velocity ($\lambda(t = 0) = 0$) and so its angular velocity $\overline{\omega}$ (see Equation (38)) is therefore initially solely tangential to the surface $\Gamma$. The No-Spin constraint is not imposed, and therefore the shell $\Sigma$ can have a non-zero spin velocity $\lambda$ during its motion as it rolls over the surface. The set of constraints provided by Equation (36) give the matrix A and the column vector b. The initial conditions are those described in the previous paragraph.*

**Example 2.** *This example is identical in all respects to Example 1, except that a non-zero initial spin velocity (i.e., a non-zero angular velocity about the normal to the surface) is now given to the shell ($\lambda(t = 0) = 3\pi$ rad/s), thereby showing the significant change in the dynamics generated by starting the shell $\Sigma$ with but a slight increase in its initial energy and a moderate initial spin velocity when compared with Example 1. All the other initial conditions are the same as those in Example 1. The set of constraints provided by Equation (36) give the matrix A and the column vector b.*

**Example 3.** *This example illustrates the results when the No-Spin constraint given in Equation (39) is added. Consistent with this constraint, the initial spin velocity is set to zero. All the other initial conditions are the same as for Example 1. The set of constraints given in Equation (42) provides the matrix A and column vector b now.*

In *each* of the three examples described above, the simulations are done for *four* different *initial* ($t = 0$) orientations of the shell $\Sigma$ relative to the surface $\Gamma$. Figure 4 shows these four initial orientations of the shell when the shell is at its initial point of contact with the surface. In what follows these four initial orientations will be referred to as Cases A to D. The lower density half of the shell (with density $d_L$) is shown in yellow with a black band; the higher density half (with density $d_H$) is shown in red. The solid (black) line shows the direction of the positive $\hat{x}$-axis of the shell in the body-fixed coordinate frame; the dash–dot (red) line shows the direction of the positive $\hat{y}$-axis, and the dotted (blue) line shows the direction of the positive $\hat{z}$-axis in the body-fixed coordinate frame.

As seen in Figure 4a, in Case A, the shell's initial orientation is such that the directions of the inertial XYZ axes coincide with the corresponding directions of the shell's body-fixed $\hat{x}\hat{y}\hat{z}$ axes. The manner in which the other three initial orientations (Cases B to D) are obtained from the orientation in Case A, and the different initial quaternion four-vectors that they engender are described in Table 1 below.

For brevity, in the simulations that follow, we show the numerical errors obtained computationally in the satisfaction of the constraints for just one case for each example. The orders of magnitude of the errors for the other cases are similar and have therefore not been shown. The computational results for each of the three examples are now presented. Animations of some of the simulations are provided in the supplements to this paper.

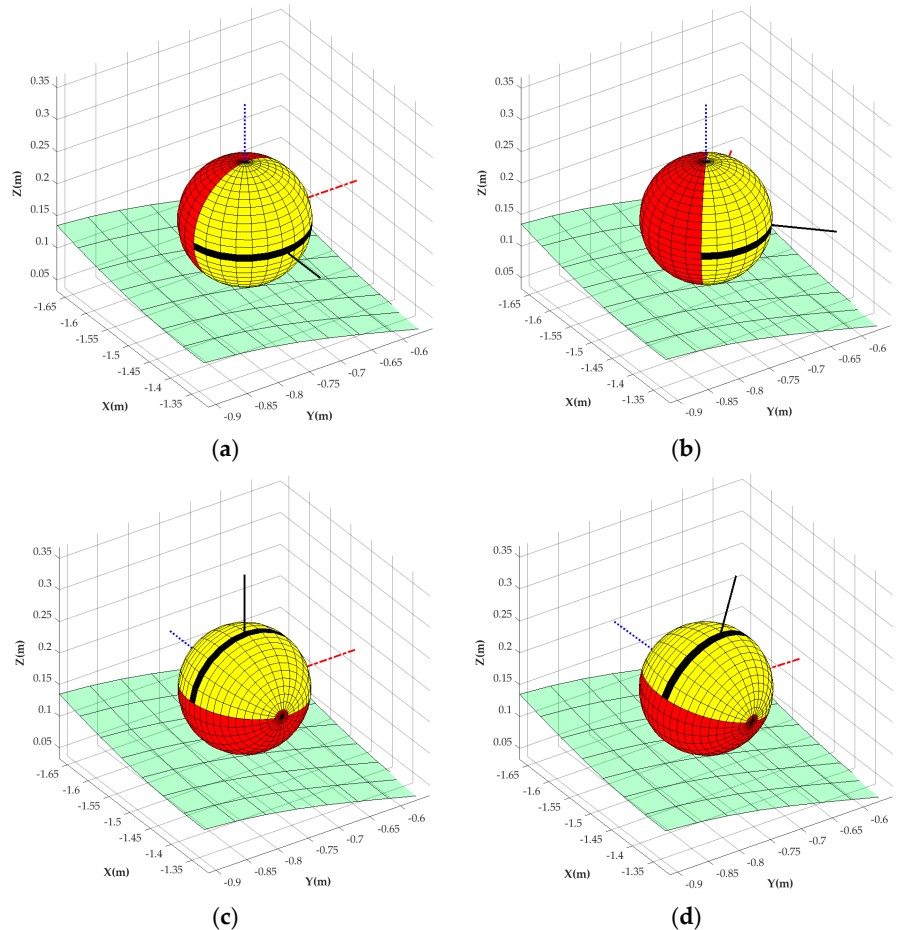

**Figure 4.** Initial orientation of the shell showing the initial orientation of the $\hat{x}\hat{y}\hat{z}$ body-fixed axis relative to the inertial XYZ frame. (**a**) Case A, (**b**) Case B, (**c**) Case C, (**d**) Case D.

**Table 1.** Initial quaternions at $t = 0$ for the initial orientations of the shell shown in Cases A–D in Figure 4.

| Case | Initial Quaternion | Initial Orientation of the Body-Fixed Frame (BFF) |
|------|--------------------|---------------------------------------------------|
| A | $[1, 0, 0, 0]^T$ | No rotation of BFF with respect to the inertial frame |
| B | $\left[\sqrt{2+\sqrt{2}}/2,\, 0,\, 0,\, \sqrt{2-\sqrt{2}}/2\right]^T$ | BFF in Case A is rotated by 45° counterclockwise around the $\hat{z}$-axis |
| C | $\left[\sqrt{2}/2,\, 0,\, -\sqrt{2}/2,\, 0\right]^T$ | BFF in Case A is rotated by 90° clockwise around the $\hat{y}$-axis |
| D | $\dfrac{1}{\sqrt{2(1+n_X)}}[1+n_X,\, 0,\, -n_Z,\, n_Y]^{T\ 1}$ | The $\hat{x}$-axis of the BFF in Case A points in the direction of the normal vector $n$ to the surface. |

$^1$ $n_X$, $n_Y$ and $n_Z$ are the three components of the unit normal vector $n(t = 0)$.

### 3.1.1. Example 1

In the first example, the constraint equations given in Equation (36) are implemented. The shell starts the motion from the four different initial orientations described in Cases A, B, C, and D with no initial spin.

Figure 5a,c,e,g show the 3D motion of the shell $\Sigma$. The solid red line shows the path traced by the point of contact $P$ on the surface $\Gamma$ as the shell rolls over it, and the dashed blue line shows the trajectory of the (geometric) center $O$ of the shell in the 3D space (see Figure 2). Figure 5b,d,f,h show these trajectories projected onto the X-Y plane for different initial orientations of the shell (Cases A to D, Figure 4) with respect to the inertial XYZ frame. As seen, a change in initial orientation even when the rest of the initial conditions are

left totally unchanged leads to dramatic changes in the path taken by the shell. Although the distance between the center of mass *C* and the center *O* of the non-uniform shell is only 0.61 cm, which is just about 7% of its outer radius, the paths taken by the shell in the four cases are significan tly different. This shows the high sensitivity of the dynamics to the *initial* starting orientation of the shell for even small deviations of its center of mass *C* from its geometrical center *O*.

Videos S1–S4 show the three-dimensional animations of the rolling motion of the shell Σ on the surface Γ described in Cases A to D, respectively. The video 'avi' files are provided in the supplementary material with the paper [33]. These animations can be seen by first downloading the (free) software called VLC and then double clicking on the downloaded files Animations of the motions shown in Figure 5a,c,e,g can be seen in this way. They have a slightly different viewing angle from those in these figures, so that the motion over the dimpled surface can be better observed. By running these animations at a slower speed than in real time, one can see the enormous complexity of the motion of the shell as it navigates across the undulating surface terrain.

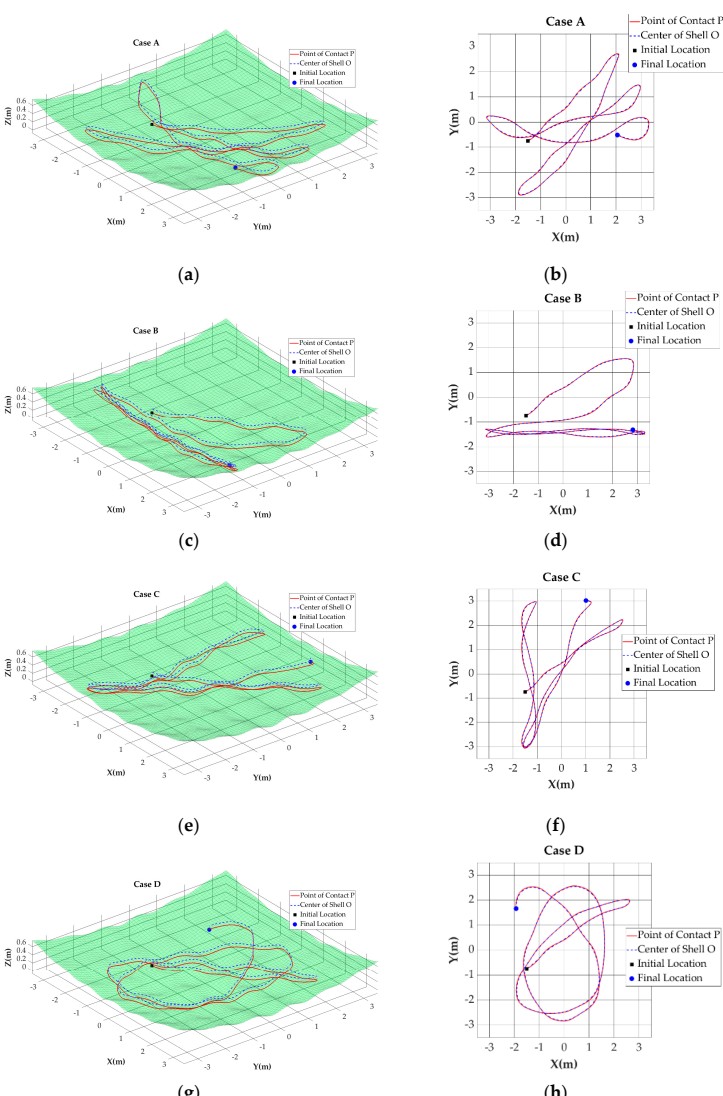

**Figure 5.** Example 1. Path taken by the point of contact *P* and the center *O* plotted in 3D space on the left and its projection on the horizontal XY plane on the right. (**a**,**b**) Case A, (**c**,**d**) Case B, (**e**,**f**) Case C, (**g**,**h**) Case D.

Figure 6a,c,e,g show the normal reaction force and the frictional force applied by the surface Γ on the shell for Cases A to D, respectively. The normal force (solid line) is positive, showing that the shell does not leave the surface during its motion. Components of the torque $T^P_{XYZ}$ provided by the surface on the shell (in the inertial XYZ coordinates) about the point of contact $P$ are shown in Figure 6b,d,f,h. They are computed using Equation (55). Since the surface meets the shell at only one point, it is impossible for the surface to apply a torque on the shell, and therefore this torque must, theoretically, be zero. As seen, this computed torque applied by the surface about the point of contact $P$ is $O(10^{-9})$ and is negligibly small. It should be noted that in this example, the shell Σ can spin freely about the normal vector $n$ to the surface at its point of contact $P$ when $t > 0$

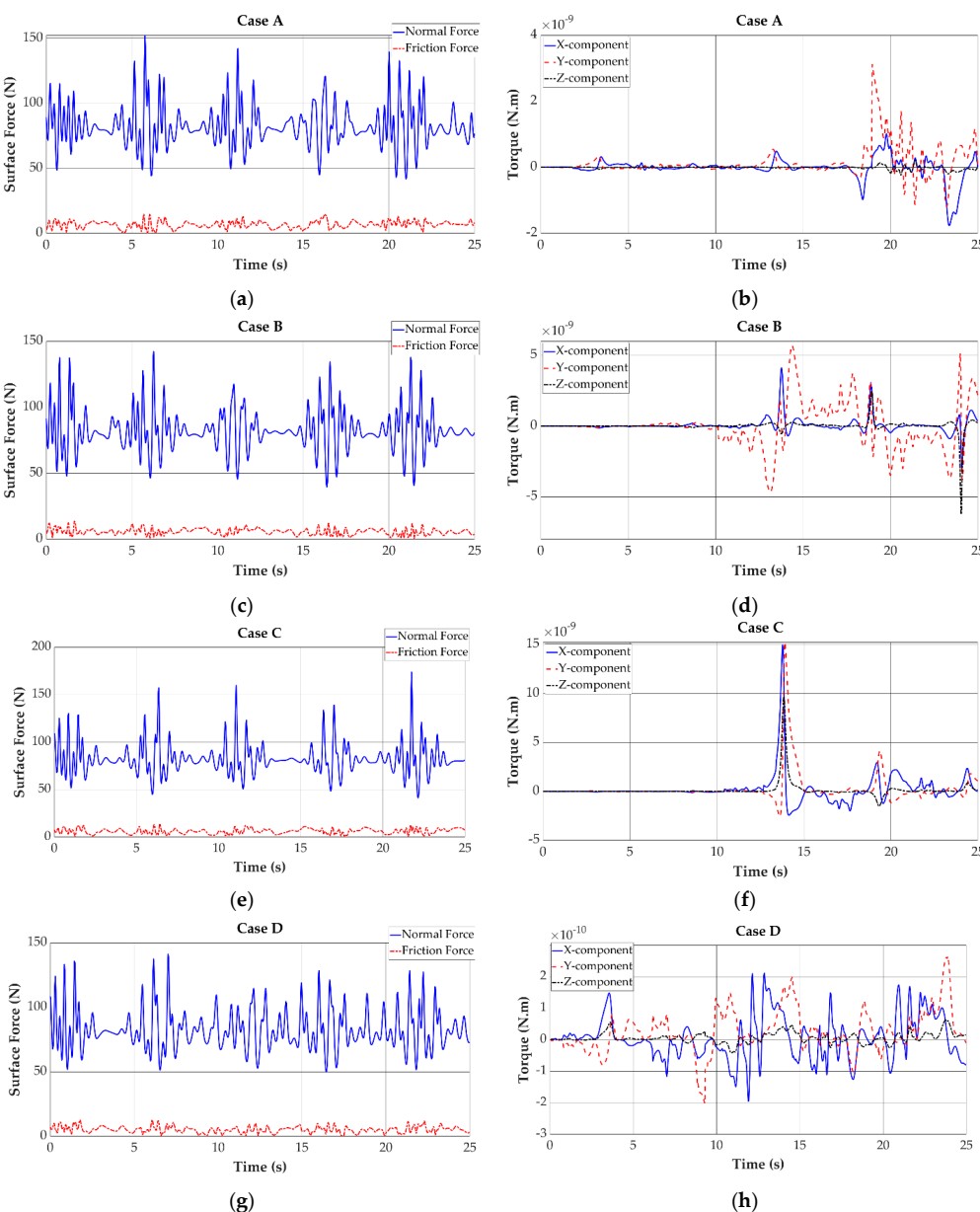

**Figure 6.** Example 1. Magnitude of normal and frictional forces provided by the surface to the shell at $P$ on the left and the computed torque $T^P_{XYZ}$ on the right. (**a**,**b**) Case A, (**c**,**d**) Case B, (**e**,**f**) Case C, (**g**,**h**) Case D.

It should be noted that the ratio of the frictional force to the normal force computed at every instant of time must be less than the coefficient of friction, $\mu$, between the surface $\Gamma$ and the shell $\Sigma$ so that the shell does not slip over the surface. At each instant of time $t$, this ratio (of the two forces) depends on the dynamics of the motion and is determined from the equations of the motion and the initial conditions; it can be obtained from the numerical results shown in Figure 6 over the duration of the shell's motion. The maximum value of this ratio therefore gives the minimum value of $\mu$ required to sustain the shell's motion without slipping. For each of the four cases, this minimum value of $\mu$ required so that the shell does not slip over the surface (over the duration of its motion) is different. We report these numbers for the four Cases A to D as: $\mu_A = 0.23$, $\mu_B = 0.19$, $\mu_C = 0.2$, and $\mu_D = 0.17$.

For Case A, the computational errors in the satisfaction of the constraints $\varphi_i(t), i = 1, \ldots, 5$. (see Equations (15), (17), (19), (22) and (27)) are shown in Figure 7. Errors in the satisfaction of the constraint $\dot{\varphi}_1$ are shown in Figure 7a. Figure 7f shows the errors in the satisfaction of the constraint $\varphi_E(t)$ normalized by the initial energy of the shell (see Equation (33)). We see that errors have orders of magnitude that are of the same order as the integration error tolerances set for the numerical integrator.

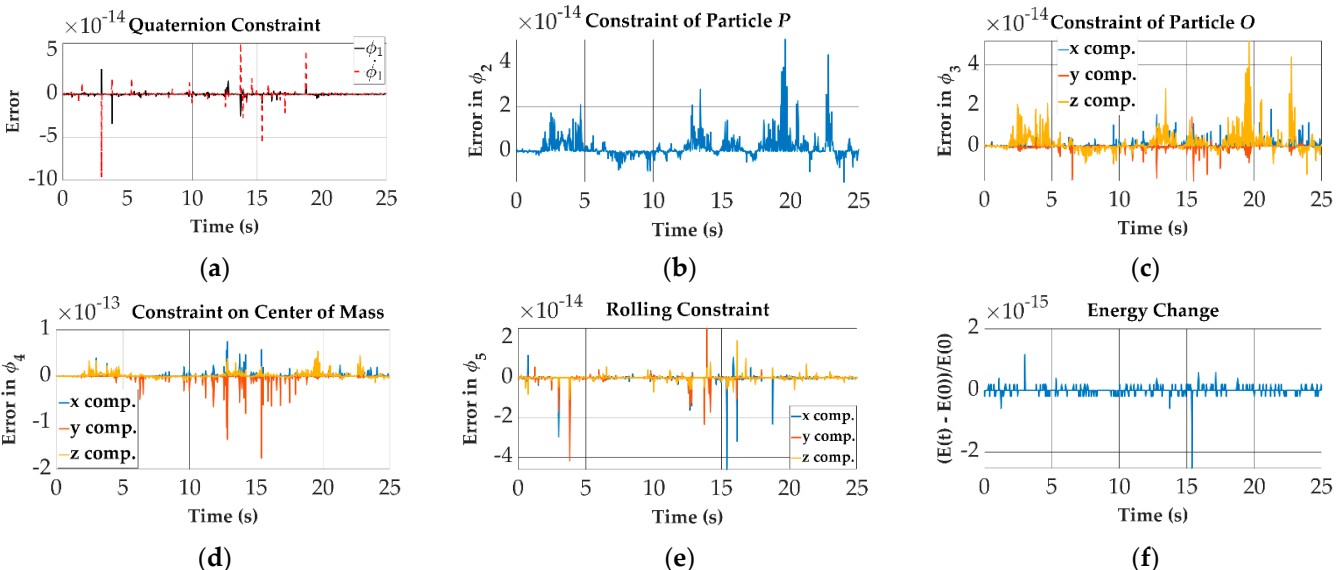

**Figure 7.** Example 1. Case A. Time history of the numerical errors in the satisfaction of the constraints (**a**) $\varphi_1$ and $\dot{\varphi}_1$, (**b**) $\varphi_2$, (**c**) $\varphi_3$, (**d**) $\varphi_4$, (**e**) $\varphi_5$, (**f**) normalized $\varphi_E$.

Each of these errors for Cases B, C, and D, have similar orders of magnitude. As mentioned earlier, for brevity, we show these errors for only one case in each example.

### 3.1.2. Example 2

In this example, the shell starts its motion from the same four initial conditions as in Example 1, except that the initial spin velocity about the unit normal vector $n$ to the surface $\Gamma$ is now nonzero. Figure 8 shows the different paths (projected on to the XY plane) taken by the point of contact $P$ (see Figure 2) on the surface $\Gamma$ during the motion for Cases A to D with and without an initial spin velocity. The solid (green) line shows the path taken by $P$ when the shell $\Sigma$ has an initial spin velocity of $\lambda(0) = 3\pi \; rad/s$, and the dashed (blue) line shows the path when the shell has no initial spin velocity.

Comparing Examples 1 and 2, the initial potential energies in the corresponding cases (Cases A to D) in the two examples are identical, and we find that the initial kinetic energies of translation in the corresponding cases are also about the same; however, the initial rotational energies in each of the cases in Example 2 are higher than the corresponding initial rotational energies in Example 1 due to the addition of the non-zero spin velocity, $\lambda(0)$. However, the total initial energies in each of the cases in Example 2 are higher than

those in the corresponding Cases A to D in Example 1 by only about 3.6%, 5.4%, 4.9%, and 5.2%.

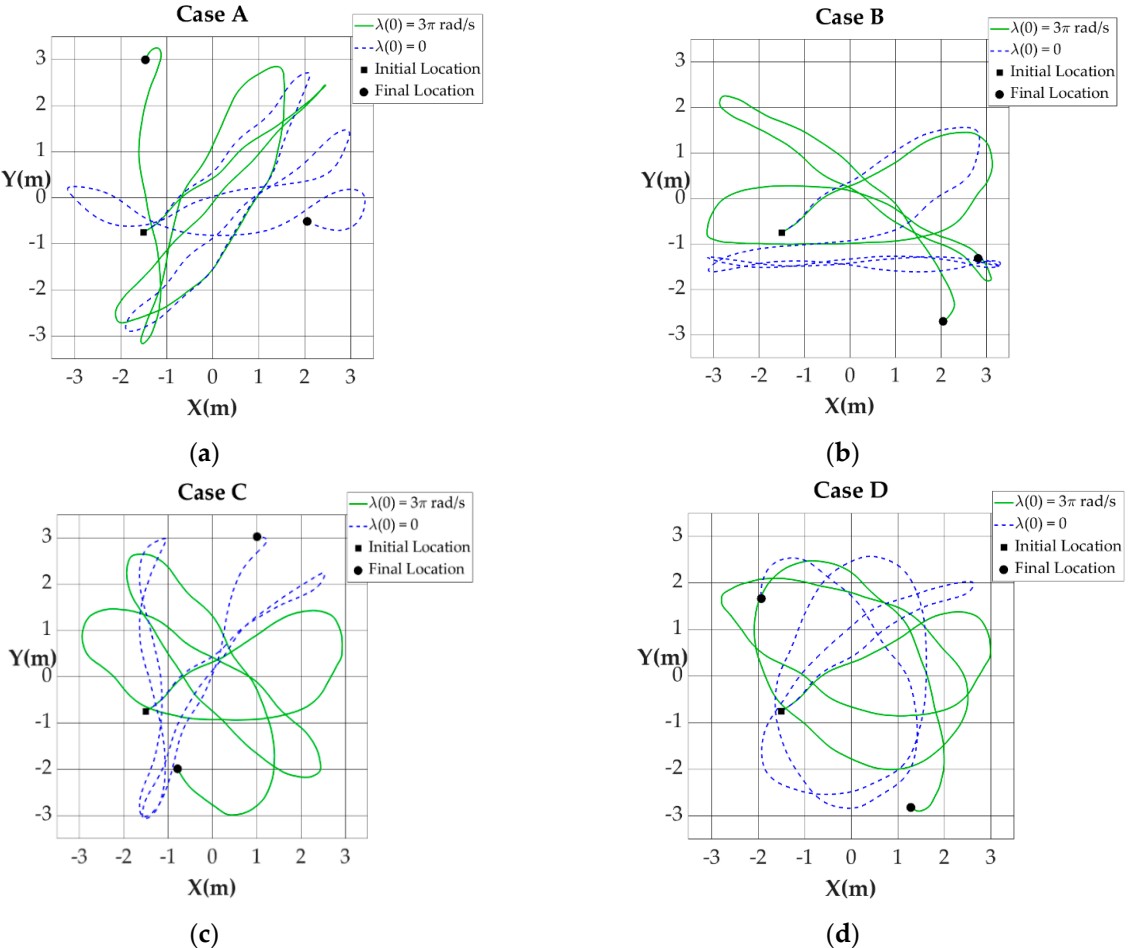

**Figure 8.** Comparison of the path taken by the point of contact $P$ projected on the $XY$ plane when $\lambda(0) = 3\pi \ rad/s$ (solid green line) and when $\lambda(0) = 0$ (dashed blue line) for (**a**) Case A, (**b**) Case B, (**c**) Case C, (**d**) Case D.

Despite this small initial energy difference between Examples 1 and 2, which is of course preserved though out the motion (as the error in satisfaction of $\varphi_E$ shows in Examples 1 and 2), as seen in Figure 8, the trajectories of the dynamical system are significantly different. We observe, then, that although the initial orientation, position, and velocity of the geometrical center are identical in the two examples, the addition of a small initial spin velocity makes a substantial change in the entire motion of the shell, and, in particular, the extent of surface territory 'explored' by the shell during its motion.

The normal reaction forces and the friction forces exerted by the surface $\Gamma$ at the point of contact between the shell and the surface are shown in Figure 9a,c,e,g for Cases A, B, C, and D, respectively. As before, the normal force is always positive showing that the shell never leaves the surface. Figure 9b,d,f,h show that for each of these cases, the computed reaction torque applied by the surface $\Gamma$ on the shell about the point of contact $P$ which, theoretically speaking should be zero as mentioned before, is computed to be of $O(10^{-9})$.

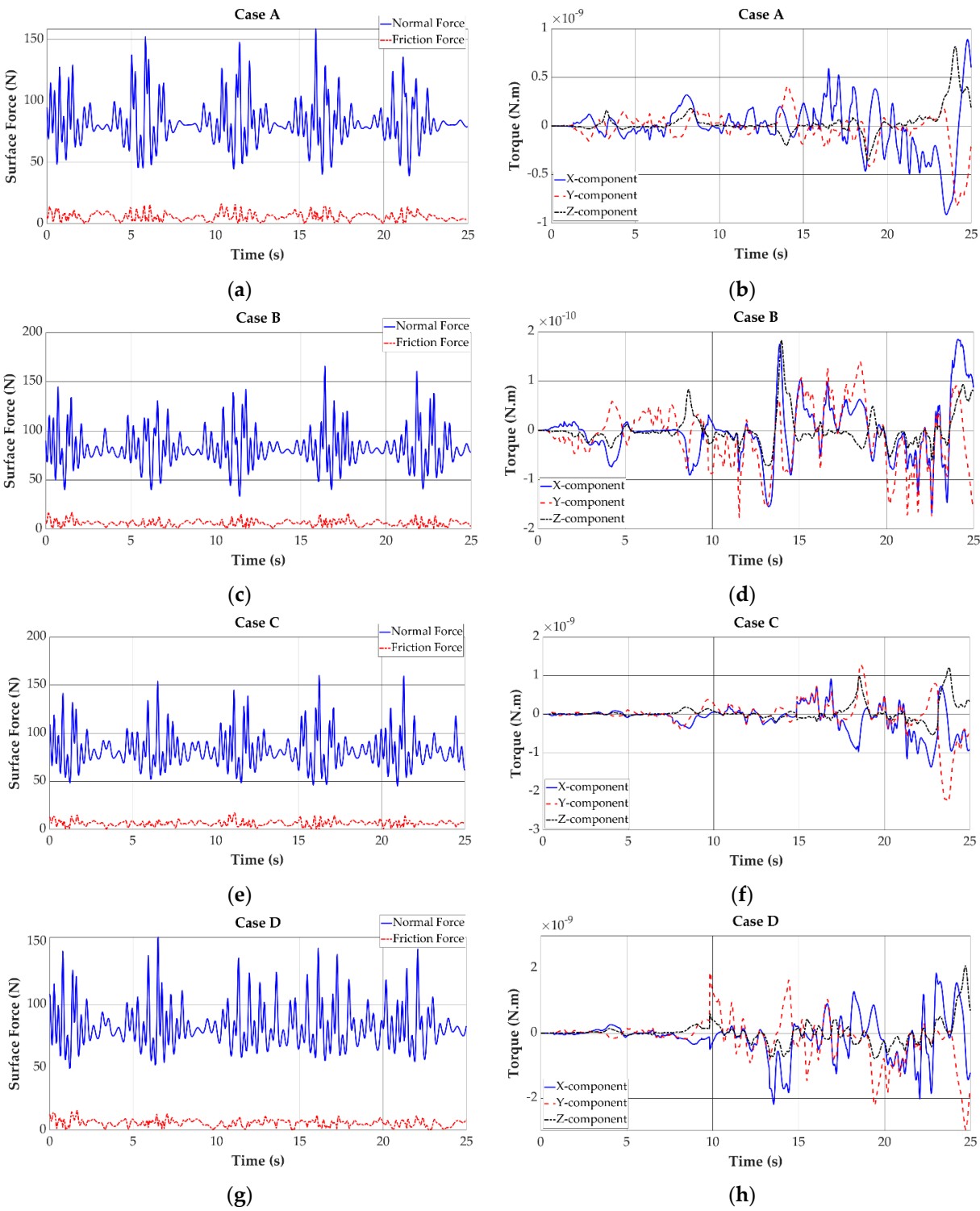

**Figure 9.** Example 2. Magnitude of normal and frictional forces provided by the surface to the shell at $P$ on the left and the computed torque $T^P_{XYZ}$ on the right. (**a**,**b**) Case A, (**c**,**d**) Case B, (**e**,**f**) Case C, (**g**,**h**) Case D.

As in Example 1, the minimum value of coefficient of friction, $\mu$, required—so that the shell $\Sigma$ does not slip over the surface $\Gamma$ throughout the duration of its computed motion—is found, as before, to be different for each case. The values for Cases A to D are, respectively,

$\mu_A = 0.26$, $\mu_B = 0.28$, $\mu_C = 0.22$ and $\mu_D = 0.19$. Comparisons with Example 1 show that the values of the coefficients of friction obtained here are higher for each of the corresponding cases A to D. This is caused by the additional rotational energy, though relatively quite small, that is provided to the shell by the initial spin velocity.

The numerical errors in the satisfaction of the constraints in Case B are shown in Figure 10. The magnitudes of these errors are seen to be of the same order as the error tolerances used for the numerical integration. Similar results are found for Cases A, C and D.

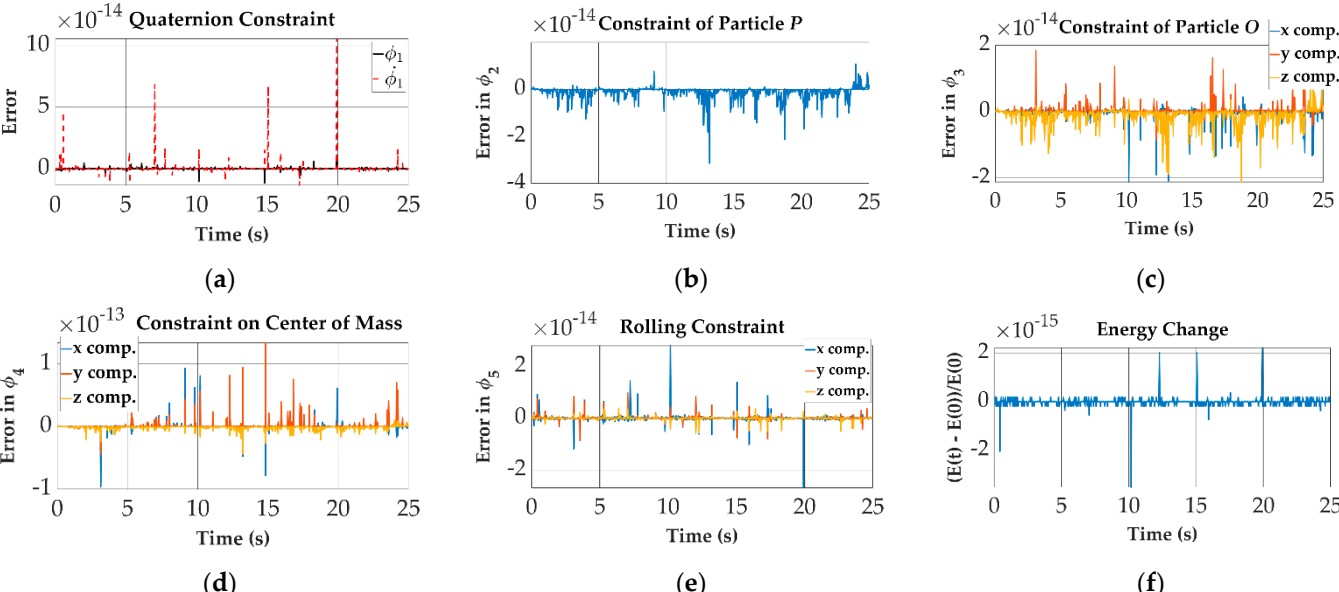

**Figure 10.** Example 2. Case B. Time history of the numerical errors in the satisfaction of the constraints (**a**) $\varphi_1$ and $\dot{\varphi}_1$, (**b**) $\varphi_2$, (**c**) $\varphi_3$, (**d**) $\varphi_4$, (**e**) $\varphi_5$, (**f**) normalized $\varphi_E$.

### 3.1.3. Example 3

In this example, the spherical shell $\Sigma$ is not permitted to spin about the normal vector to the surface $\Gamma$ during its motion, and the No-Spin constraint $\varphi_{NS}$ given in Equation (39) is enforced. The only difference between the simulation for this example and the simulation for Example 1 is the addition of this No-Spin constraint. The initial total energy in each case (for Cases A–D) is therefore the same as the initial total energy for the corresponding case in Example 1. The matrix $A$ and the vector $b$ that describe the constraint set are now given by Equation (42).

Figure 11 compares the path taken by the point of contact $P$ on the surface $\Gamma$ obtained in this example for Cases A, B, C, and D with the corresponding path computed in Example 1. The solid (green) line shows the shell's motion when the No-Spin constraint is enforced, and the dashed (blue) line shows the motion obtained in Example 1 when this constraint is not enforced. We observe that the presence of this additional constraint makes a substantial change in the dynamical behavior of the shell. Though the two sets of lines for each case start at the same point, their paths soon become markedly different for each case, as seen in this figure.

Figure 12a,c,e,g show the reaction force supplied by the surface at $P$ for Cases A, B, C, and D, respectively. As before, the reaction force normal to the surface is always positive in all four cases showing that the shell does not leave the surface.

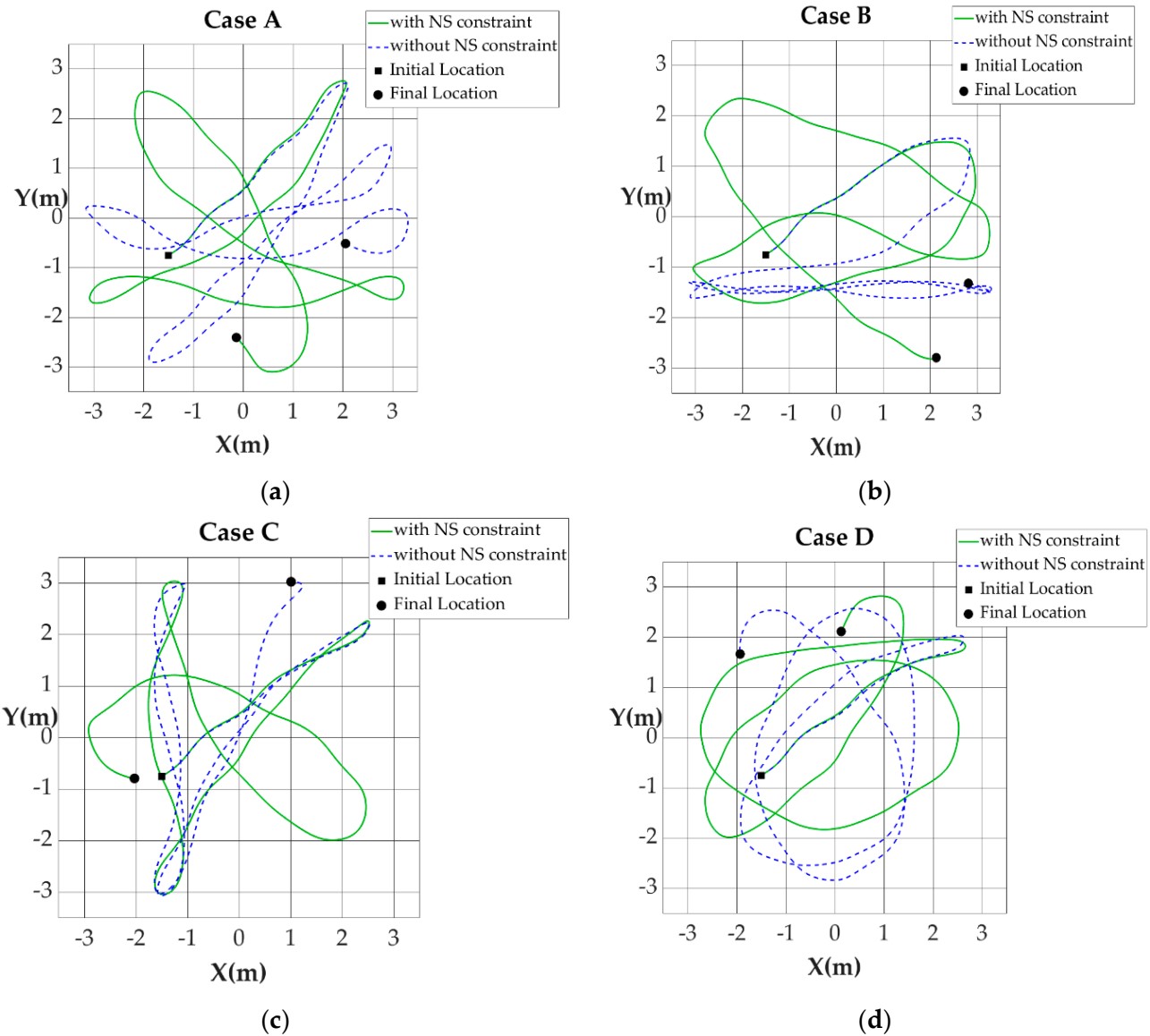

**Figure 11.** Comparison of the path taken by the point of contact *P* projected on the XY plane. The solid (green) line shows the trajectory with the No-Spin (NS) constraint and the dashed (blue) line without the No-Spin (NS) constraint for (**a**) Case A, (**b**) Case B, (**c**) Case C, (**d**) Case D.

Figure 12b,d,f,h show the components of the torque, $T^P_{XYZ}$, normal to the surface $\Gamma$ and tangential to it. As seen, the component tangential to the surface is negligibly small for each of the Cases A to D. However, the component of the torque normal to the surface is seen to be substantial for each case, unlike in Examples 1 and 2, both of which show that the X, Y, and Z components of the torque are all negligibly small (see Figure 6 and Figure 9 for comparison). Since the shell $\Sigma$ meets the surface $\Gamma$ at only a single point, the surface cannot apply any torque on the shell. Thus, this *additional* torque about the point *P* along the normal *n* to the surface needs to be *externally applied* to the shell at each instant of time in order to prevent it from having any angular velocity about the normal to the surface $\Gamma$, which is, of course, the No-Spin constraint under which the shell has been modeled to move in this example.

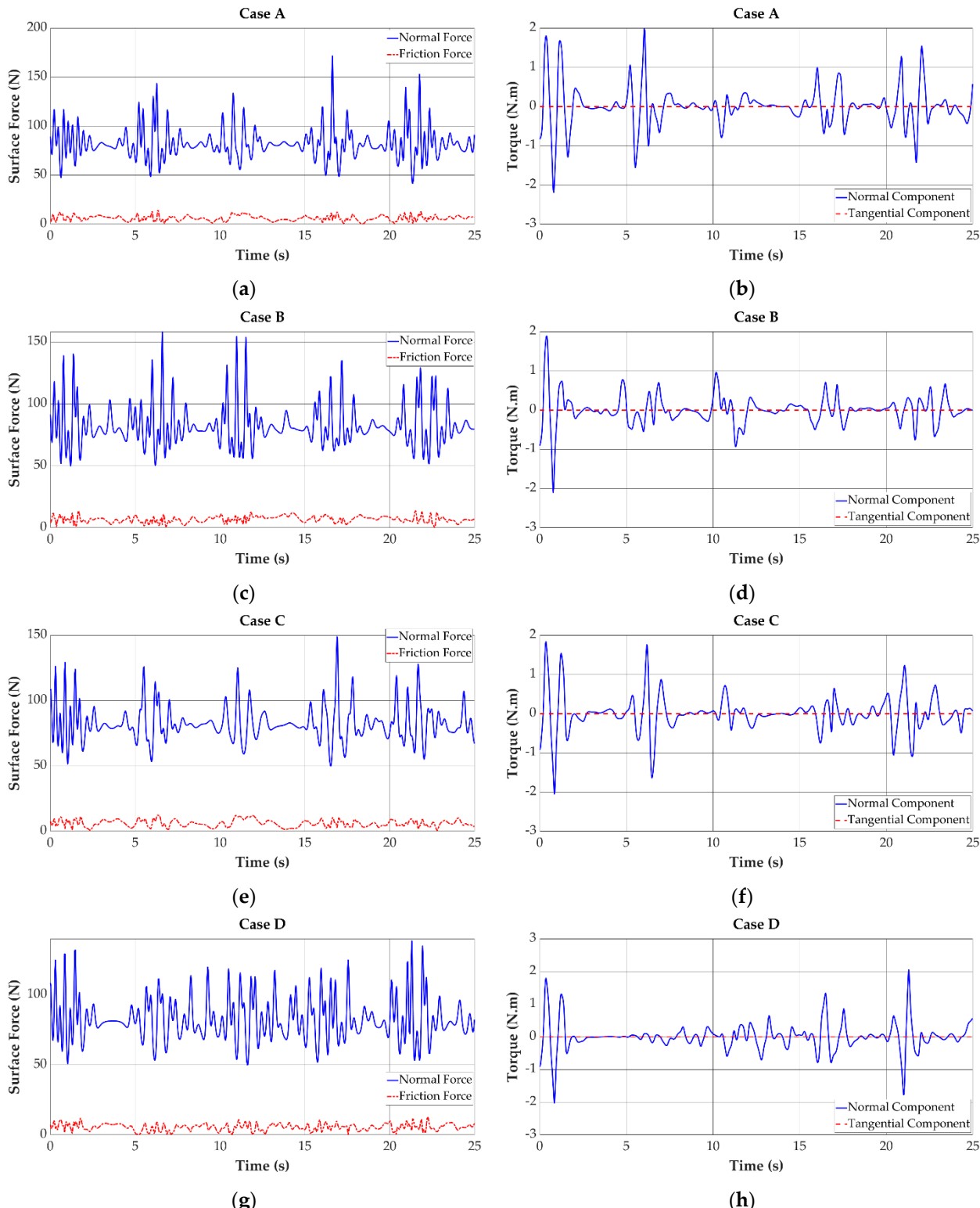

**Figure 12.** Example 3. Magnitude of normal and frictional forces provided by the surface to the shell at $P$ on the left and the torque $T^P_{XYZ}$ that is required to be *externally applied* to satisfy the No-Spin constraint on the right. (**a**,**b**) Case A, (**c**,**d**) Case B, (**e**,**f**) Case C, (**g**,**h**) Case D.

The minimum values of the coefficient of friction required for the shell $\Sigma$ not to slip over the surface $\Gamma$ during its computed motion for Cases A to D are: $\mu_A = 0.21$, $\mu_B = 0.19$, $\mu_C = 0.16$ and $\mu_D = 0.17$.

The numerical errors in the satisfaction of the constraints for Case C are shown in Figure 13. Their magnitudes are of the same order as the error tolerances used in the numerical integration as before. The time history of the errors in the satisfaction of the No-Spin constraint $\varphi_{NS}$ are also shown (see Equation (39)) in this figure. Similar results are found for Cases A, B and D.

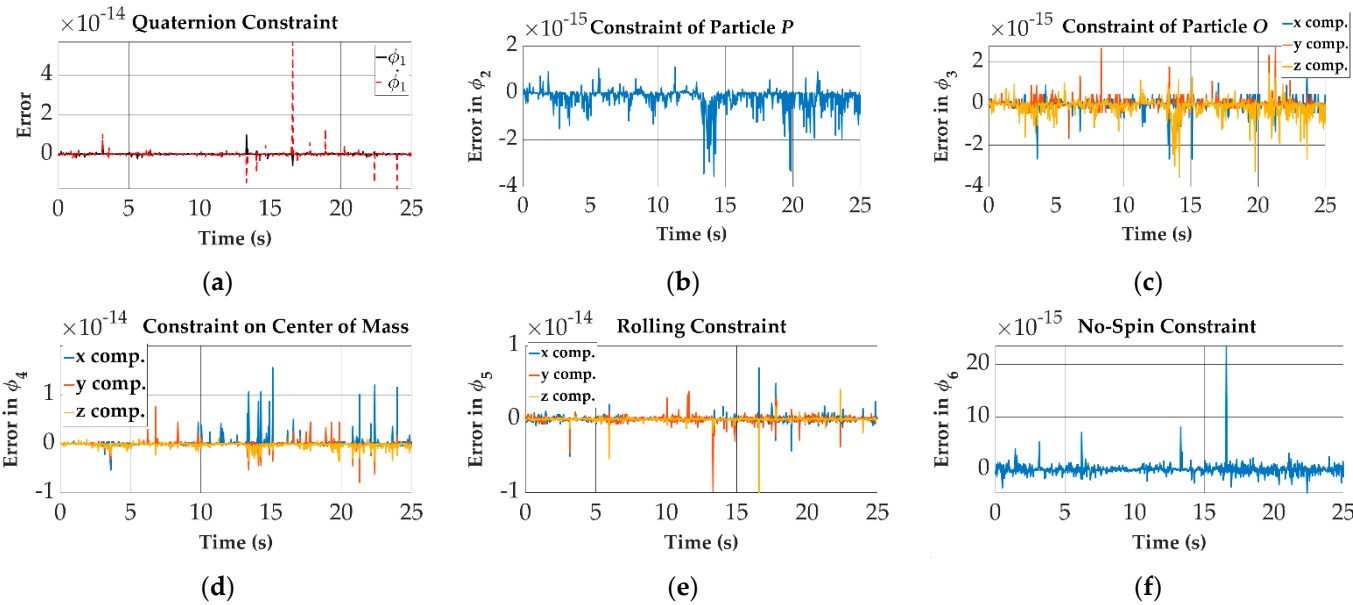

**Figure 13.** Example 3. Case C. Time history of the numerical errors in the satisfaction of the constraints (**a**) $\varphi_1$ and $\dot{\varphi}_1$, (**b**) $\varphi_2$, (**c**) $\varphi_3$, (**d**) $\varphi_4$, (**e**) $\varphi_5$, (**f**) normalized $\varphi_E$.

We can now compare among the three examples the coefficients of friction that have been reported in order to sustain the motion of the shell without slipping over the surface described by Equation (57). In Cases A to D, for the initial conditions chosen, the coefficients reported when the shell is given an initial angular velocity about the normal to the surface (Example 2) appear to be the larger than (or equal to) the corresponding coefficients reported in the other two examples. In addition, the coefficients of friction obtained when the No-Spin constraint is enforced (Example 3) appear to be lower than (or equal to), the corresponding coefficients of friction for each of the cases reported in Examples 1 and 2.

## 4. Discussion

Since the approach developed in this paper is new, we provide in this section: (1) a discussion of the general approach developed here for analytically and computationally determining the dynamical behavior of complex mechanical systems, and (2) a discussion of the simulation results obtained by using the new approach to the open problem of the shell rolling, without slipping, on an arbitrarily prescribed surface.

(1) This paper points to new directions in the conceptualization, analysis, and computational aspects of complex multi-body dynamical systems. In general, there is a minimum number of variables that can completely describe all the configurations of a dynamical system. However, describing a dynamical system by using a number of coordinates (variables) that is greater than the minimum adequate number makes a notable advance in the modeling and computer simulation of systems. This capability facilitates the description and analysis of the dynamics of complex systems since the coordinate vector is not restricted to include a certain number of independent coordinates. Often, one finds a set of coordinates that play crucial roles in the description and understanding of the dynamics of a system, though the coordinates in the set may not all be independent. Their explicit involvement in the equations of motion then provides an improved understanding of the system's dynamics. It improves our capability to algebraically and geometrically appreciate the sensitivity in the evolving dynamics of these coordinates to different initial conditions and

parameter changes. For instance, as shown in Section 3, the tracks made on an undulating surface by a non-uniform spherical shell as it rolls on it show great sensitivity to the initial orientation of the shell and could be of vital significance in its navigation over the surface, so that the coordinate of the shell's point of contact with the surface becomes important both from a geometric and a physical viewpoint.

The novel new direction in the modeling of complex dynamical systems presented in this paper expands the coordinate vector of a primary mechanical system by introducing zero-mass particles. The primary mechanical system in addition to the zero-mass particles then forms a multibody system. The zero-mass particles, as components of the multibody system, are constrained to co-locate with intended points in the system whose motions provide crucial and important understandings of the system's complex dynamics. The additional coordinates of the zero-mass particles become intrinsic to the multibody system. They are involved in, and used, in the derivation of the equations of motion of the multibody system, and their motion is explicitly obtained as the system evolves in time. These additional coordinates, however, are not independent and one has to invoke the theory of constrained motion to obtain the correct equations of motion of such multibody systems.

We note that particles with zero mass have so far never been used in nonrelativistic mechanics. From a Newtonian standpoint, no forces can be applied to zero-mass particles, and therefore their accelerations are indeterminate. The new methodology developed here utilizes zero-mass particles within the context of the dynamics of constrained motion. These developments are applicable to the systems with semi-definite mass matrices. Thus, although Newton's second law of motion gives a trivial equation for a particle with zero mass (0 = 0!), well-defined constraint equations can permit the acceleration of such a particle to be uniquely determined.

The first step in the modeling methodology developed in Section 2 is to write the equation of the unconstrained system. In such a hypothetical system, all the coordinates are assumed to be independent. In the problem of the rolling sphere, for instance, the unconstrained system consists of a rigid spherical shell moving under the force of gravity and the two zero-mass particles. The equations of motion of the zero-mass particles are trivial and they make the mass matrix of this unconstrained system singular. The use of quaternions to model rotational dynamics also leads to singular mass matrices, as shown in Equation (8). The equation of the unconstrained system is stated, using generalized coordinates, in form $M\ddot{q} = Q$, where the matrix $M$ is singular.

The second step is to write the system of constraint equations. The set of constraint equations include those on the motion of the zero-mass particles and can be made to assume the form $A\ddot{q} = b$. The accelerations of the system, including the zero-mass particles, are fully defined by the constraint equations and the equations of motion of the unconstrained system if the matrix $\left[ M \,\big|\, A^T \right]^T$ has full column rank. In this paper, this is proved analytically for the example problem of a rolling spherical shell on an arbitrary surface.

The third and final step is to find the unique acceleration of the system. The matrices $M$ and $A$ and the vectors $Q$ and $b$, obtained in the preceding steps, are used to obtain the closed form analytical equations of motion for the mechanical system. This three-step procedure allows the constraint and/or reaction forces to be obtained explicitly also.

(2) The general approach explained above is applied to solve a well-known non-holonomic system that has been investigated for decades, though with little resolution to date: the problem of a rigid non-uniform spherical shell rolling over an arbitrary surface under the force of gravity. Noting that two critical points in the system—the location of the point of contact of the shell with the surface, and the geometric center of the shell—deliver considerable information about the evolving dynamics, a multi-body system comprising the shell and two zero-mass particles is considered. The two zero-mass particles are collocated with these two critical points of the system that play a significant role in describing the dynamics of the rolling sphere. Their use greatly facilitates the derivation of the explicit equations of motion of the constrained motion. Furthermore, these two points play key

roles in simplifying the description of the geometry of the constraints under which the rolling, without slipping, occurs.

In the first example the shell starts at an initial location on the bowl, with its center being given an initial velocity. In the second, an additional initial spin velocity (an angular velocity about the normal to the surface) is imparted to the shell. The third example is the same as the first, except that now the shell is prevented from rotating about the normal to the surface as it rolls over it. For each of these examples the computations investigate the influence of four different initial orientations of the shell (relative to the bowl-shaped surface) on its subsequent motion. It is shown that the motion of the shell—the track it makes on the surface as it rolls *and* the regional extent of the surface that it traverses—is very sensitive to its initial orientation. This is a consequence of its non-uniformity, though this non-uniformity as measured by the distance of its center of mass from its geometric center is indeed very small, a somewhat non-intuitive result. Furthermore, it is shown that the presence of an initial spin velocity given to the shell about the normal to the surface substantially affects the shell's dynamical behavior and heavily influences the region of its traverse. It seems to call for larger coefficients of friction between the shell and the surface to keep the shell from slipping, as one might intuitively expect. Preventing the shell from rotating about the normal to the surface provides an additional constraint, which the approach developed here is shown to easily handle. The inclusion of this constraint also shows significant changes in the dynamical behavior. The minimum coefficients of friction required to sustain the motion of the shell without it slipping over the surface are determined for each example and each of the initial orientations considered. Among the three examples, the minimum coefficients of friction required are found to be the largest when the shell is given an initial spin velocity and the least when the shell is prevented from rotating about the normal to the surface. The analytically obtained (generalized) reaction forces generated by the surface as the shell rolls over it show wide variations depending on the local acceleration and the local curvature of the surface. It should be noted that due to the complexity of the dynamics and the arbitrariness of the surface, at the present time, we are precluded for inducing general statements about the shell's motion other than the presence of some simple invariants such as its mechanical energy. The complexity of the dynamics can perhaps best be intuitively grasped by viewing Videos S1–S4 provided in the supplementary material for this paper [33].

## 5. Conclusions

This paper explores a new direction in the modeling and simulation of complex mechanical systems. The key idea that has been presented is the use of particles of zero mass that can be co-located with points in the system whose motion is critical to our understanding of its dynamical behavior. The coordinates of such points are included in the mathematical modeling of the system, which in turn is facilitated by their presence. This could be not determined in the past, since particles of zero mass cannot be subjected to any forces. Their inclusion in the mass matrix of any mechanical system makes the matrix singular, therefore the accelerations of any system in which they are included cannot be obtained. What makes it possible to use such zero-mass particles is the change in our perspective brought about by viewing the system as a system subjected to constraints. The paper shows how the explicit equations of motion of a constrained system that includes zero-mass particles can be obtained. This also allows the motion of those critical points in a system to which the zero-mass particles are attached to be handily determined.

The classical problem of obtaining the explicit equations of motion for a non-uniform spherical shell rolling, without slipping, was first broached in 1895. The determination of the explicit equations of motion for a spherical shell moving over an arbitrarily prescribed surface has remained an open problem for more than a century now. The new approach presented here is applied to handle this problem. It places two particles of zero-mass, one at the geometric center of the shell, the other at the point of contact between the shell and the surface. The paper shows how to obtain the explicit equations of motion and obtains

them. This allows the motion of the shell to be simulated in a straightforward manner. The simulations show that the motions are extremely complex and highly sensitive to parameters such as the initial orientation of the shell, its initial conditions, and the nature of the surface on which it moves. The complexity of the equations of motion of a shell as it moves over an arbitrarily prescribed surface, however, precludes any general conclusions from being drawn from the simulations at this time, due to limitations in the current state of knowledge of highly nonlinear multi-degree of freedom mechanical systems.

**Supplementary Materials:** Video S1: Example 1, Case A: https://doi.org/10.5281/zenodo.6872914. Video S2: Example 1, Case B: https://doi.org/10.5281/zenodo.6872914. Video S3: Example 1, Case C: https://doi.org/10.5281/zenodo.6872914. Video S4: Example 1, Case D: https://doi.org/10.5281/zenodo.6872914

**Author Contributions:** Conceptualization, F.E.U. and N.M.; methodology, F.E.U.; software, F.E.U. and N.M.; computation, N.M. and F.E.U.; formal analysis, F.E.U. and N.M.; writing, F.E.U.; review and editing, F.E.U.; visualization, N.M. and F.E.U.; supervision, F.E.U. All authors have read and agreed to the published version of the manuscript.

**Funding:** The research received no external funding.

**Data Availability Statement:** The data presented in this study are available from the corresponding author upon request.

**Conflicts of Interest:** The authors declare no conflict of interest.

**Appendix A**

In this appendix, the matrices $A_3$ and $b_3$ are derived. The second time derivative of Equation (19) can be written as

$$\ddot{\beta} - \ddot{\alpha} - r\ddot{n} = 0. \tag{A1}$$

Taking the time derivatives of unit vector $n$ yields

$$
\begin{aligned}
\dot{n} &= \frac{\dot{k}}{\sqrt{k^T k}} - \frac{k^T \dot{k}}{\left(k^T k\right)^{3/2}} k \\
&= \frac{1}{\sqrt{k^T k}} \dot{k} - \frac{kk^T}{\left(k^T k\right)^{3/2}} \dot{k} \left[ \frac{1}{\sqrt{k^T k}} I_3 - \frac{kk^T}{\left(k^T k\right)^{3/2}} \right] \frac{\partial k}{\partial \alpha} \dot{\alpha} \\
&= \Delta \frac{\partial k}{\partial \alpha} \dot{\alpha}.
\end{aligned}
\tag{A2}
$$

Differentiating once more, after some algebra, we obtain

$$\ddot{n} = \Delta \frac{\partial k}{\partial \alpha} \ddot{\alpha} + \Delta \frac{d}{dt}\left(\frac{\partial k}{\partial \alpha}\right)\dot{\alpha} + \delta k - \frac{2\dot{k}^T k}{\left(k^T k\right)^{3/2}} \dot{k} \tag{A3}$$

where

$$\Delta = \frac{1}{\sqrt{k^T k}}\left[ I_3 - \frac{kk^T}{k^T k} \right], \quad \delta = \frac{1}{\left(k^T k\right)^{3/2}}\left[ \frac{3\left(\dot{k}^T k\right)^2}{k^T k} - \dot{k}^T \dot{k} \right] \text{ and } \dot{k} = \frac{\partial k}{\partial \alpha}\dot{\alpha}. \tag{A4}$$

Furthermore, from Equation (19), which is $\beta = \alpha + rn := f(\alpha)$, the Jacobian, $J_f$, of $f(\alpha)$ is obtained as

$$\dot{\beta} = \dot{\alpha} + r\dot{n} = \left[ I_{3\times3} + r\Delta\frac{\partial k}{\partial \alpha} \right]\dot{\alpha} := J_f \dot{\alpha}. \tag{A5}$$

The second equality follows from Equation (A2).
From Equation (A1), the constraint equation can then be written as

$$A_3 \ddot{q} := \begin{bmatrix} 0_{3 \times 3} & | & 0_{3 \times 4} & | & I_{3 \times 3} + r \Delta \frac{\partial k}{\partial \alpha} & | & -I_{3 \times 3} \end{bmatrix} \ddot{q} = -r \left[ \Delta \frac{d}{dt} \left( \frac{\partial k}{\partial \alpha} \right) \dot{\alpha} + k \delta - \frac{2 \dot{k}^T k}{\left( k^T k \right)^{3/2}} \dot{k} \right] := b_3. \tag{A6}$$

so that

$$A_3 = \begin{bmatrix} 0_{3 \times 3} & | & 0_{3 \times 4} & | & J_f & | & -I_{3 \times 3} \end{bmatrix}, \text{ and } b_3 = -r \left[ \Delta \frac{d}{dt} \left( \frac{\partial k}{\partial \alpha} \right) \dot{\alpha} + k \delta - \frac{2 \dot{k}^T k}{\left( k^T k \right)^{3/2}} \frac{\partial k}{\partial \alpha} \dot{\alpha} \right]. \tag{A7}$$

**Appendix B**

To obtain the angular velocity of the rolling shell in terms of the velocity of its geometrical center and the spin velocity around the unit normal vector, we consider the vector of angular velocity whose components are measured in the inertial frame defined as

$$\overline{\omega} := \begin{bmatrix} \omega_X \\ \omega_Y \\ \omega_Z \end{bmatrix} = S \omega, \tag{A8}$$

where $\omega$ is the angular velocity with components expressed in the body-fixed coordinate frame $\hat{x} \hat{y} \hat{z}$, and the three by three matrix $S$ is given in Equation (23).

The rolling constraint stated in Equation (27) yields the equation

$$\widetilde{n} \overline{\omega} = -\frac{\dot{\beta}}{r} \tag{A9}$$

that relates the angular velocity of the shell and the linear velocity of its geometrical center, $O$. The three by three coefficient matrix $\widetilde{n}$ is the skew-symmetric matrix given in Equation (28), which is singular. Recall, the unit normal to the surface $\Gamma$ at the point $P$ is $n = \begin{bmatrix} n_X & n_Y & n_Z \end{bmatrix}^T$.

The solution $\overline{\omega}$ of the linear Equation (A9) is easily obtained using the Moore–Penrose (MP) generalized inverse as

$$\overline{\omega} = -(\widetilde{n})^+ \left( \frac{\dot{\beta}}{r} \right) + \left( I - (\widetilde{n})^+ \widetilde{n} \right) \hat{\lambda} \tag{A10}$$

where $(\widetilde{n})^+$ is the MP-inverse of $\widetilde{n}$ and $\hat{\lambda}$ is an arbitrary three-vector. We observe that the second member on the right in Equation (A10) belongs to the null space of the matrix $\widetilde{n}$.

Since $\| n \|^2 = n_X{}^2 + n_Y{}^2 + n_Z{}^2 = 1$ and $(\widetilde{n})^+ = -\widetilde{n}$, we find that

$$I - (\widetilde{n})^+ \widetilde{n} = I - \begin{bmatrix} 0 & n_Z & -n_Y \\ -n_Z & 0 & n_X \\ n_Y & -n_X & 0 \end{bmatrix} \begin{bmatrix} 0 & -n_Z & n_Y \\ n_Z & 0 & -n_X \\ -n_Y & n_X & 0 \end{bmatrix} = \begin{bmatrix} n_X \\ n_Y \\ n_Z \end{bmatrix} \begin{bmatrix} n_X & n_Y & n_Z \end{bmatrix} = n n^T. \tag{A11}$$

Thus, from Equation (A10) the angular velocity $\overline{\omega}$ is

$$\overline{\omega} = \widetilde{n} \frac{\dot{\beta}}{r} + n \underbrace{n^T \hat{\lambda}}_{=\lambda} = \widetilde{n} \frac{\dot{\beta}}{r} + \lambda n. \tag{A12}$$

where, as seen from the second member in the second equality above, the scalar $\lambda = n^T \hat{\lambda}$ gives the component of the angular velocity along the unit normal vector $n$ to the surface $\Gamma$.

Using Equation (A8), we then obtain

$$S \omega = \widetilde{n} \frac{\dot{\beta}}{r} + \lambda n. \tag{A13}$$

The angular velocity component, $\lambda$, along the unit normal $n$ can be obtained by pre-multiplying both sides of Equation (A13) by $n^T$. Noting that $n^T \tilde{n} = \left( \tilde{n}^T n \right)^T = -\left( \tilde{n} n \right)^T = 0$ and $n^T n = 1$, we obtain $\lambda = n^T S \omega$. We call $\lambda$ the 'spin velocity' of the shell $S$. When the spin velocity is zero $\overline{\omega}$ $(\omega)$ depends only on the velocity $\dot{\beta}$ of the center, $O$, of the shell whose outer radius is $r$.

Equation (A13) shows the decomposition of the angular velocity in two orthogonal components since $n^T \tilde{n} = 0$; the first member on the right-hand side of the equation is the component tangential to the surface $\Gamma$ and the second member is the component normal to the surface.

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
