# Peer review of "New Directions in Modeling and Computational Methods for Complex Mechanical Dynamical Systems"

_processes, doi:10.3390/pr10081560_

Round 1

Reviewer 2 Report

Thanks for the interesting article. I have a few questions:

1. Can a complex multiparticle dynamical system consist of smaller micron-sized particles?

2. Energy dissipation is not taken into account. Could you write the reason?

3. You mentioned „The new conceptualization uses the idea of including particles of zero mass to describe the dynamics of such systems“.  You gave an explanation, but could you give more explanation, because still without mass it is more for kinematics than for dynamics.

4. If possible, could you please bold the letters that represent vectors in equations.

Reviewer 3 Report

Journal: Processes (ISSN 2227-9717)

Manuscript ID: processes-1820969

Title: New Directions in Modeling and Computational Methods for Complex Mechanical Dynamical Systems

In this paper, the authors studied a new conceptualization of complex nonlinear mechanical systems and develops new and novel computational methods for determining their response to given applied forces and torques. The new conceptualization uses the idea of including particles of zero mass to describe the dynamics of such systems. The computations illustrate the complexity of the dynamical behavior of the system and its high sensitivity to the initial orientation of the shell and to the presence of any initial angular velocity normal to the surface.

The research is of great interest. The following manuscript has some weaknesses and here are the most important topics/questions to be dealt with:

1. Both the authors are from the University of Southern California, but use a personal email for this manuscript. Could you please use the institute email address instead?

2. Please rewrite the abstract by identifying the purpose, the problem, the methodology and the important results (not all) and conclusions of your work.

3. Introduction

The Introduction should consist of five paragraphs answering the following five questions:

What is the problem?

Why is it interesting and important?

Why is it hard?

Why hasn't it been solved before? (Or, what's wrong with previously proposed solutions?)

What are the key components of my approach and results? 

As the above-mentioned questions should be replied to, it will be better that write more relation to the benefits & disadvantages of the blending techniques and more investigate about limitations of previous studies. Also, this part needs more explanations to state clearly the objectives & hypothesis of this study at the end of the Introduction part. It should be mentioned to the factors that be shed light by this study.

4. Conclusion section is extremely long. The conclusions are very weak and ít requires a deeper analysis of the results.

5. What is the next step in the research activity? Please provide some information about further work in the Conclusions section.

The reviewer suggests carefully reading the whole manuscript again before resubmitting it to the journal Energies. Authors should consider the above-mentioned remarks in order to revise the manuscript. The reviewer thinks that a publication of the draft manuscript may be possible after a minor revision.

Round 2

Reviewer 1 Report

The authors have included almost all my comments and suggestions. Now, we recommend this paper for possible publication in Processes.